# Long-term sea level rise modeling of a basin-tidal inlet system reveals sediment sinks

Kevin C. Hanegan [1], Duncan M. FitzGerald [2], Ioannis Y. Georgiou [3] ✉ & Zoe J. Hughes [2]

Much of the world's population lives close to coastlines and this proximity is becoming increasingly impactful because of sea-level rise (SLR). Barrier islands and backbarrier saltmarshes, which comprise >10% of these coasts, are particularly susceptible. To better understand this risk, we model backbarrier morphologic and hydrodynamic evolution over a 200-year period of SLR, incorporating an erodible bed and a range of grain sizes. Here, we show that reduction in intertidal area creates negative feedback, shifting transport of coarse sediment (silt and sand) through the inlet from net export to net import. Imposing a modest marsh vertical accretion rate decreases the period of silt and sand import to 40 years (years 90 to 130) before being exported again. Clay is continuously exported thereby decreasing inorganic deposition on marshes and threatening their sustainability. Simulated marsh loss increases tidal prism and the volume of sand contained in ebb deltas, depleting coastal sand resources.

Quantifying the physical response of coastal systems to SLR remains one of the most important tasks within the fields of coastal geology and engineering. During the 20th century, global (eustatic) Mean Sea Level (GMSL) rose at a rate of ~2 mm/yr, but this rate has steadily increased, and by 2100, projections suggest a cumulative rise of between 0.79 and 1.46 m (RCP 8.5)[1]. SLR increases the risk of coastal flooding and storm impacts, particularly for the ~10% of the global population who live in coastal areas less than 10 m above sea level[2]. Continued coastal population growth[3], accelerating sea-level rise[4], and increased storminess[5] exacerbate this threat to coastal inhabitants, infrastructure, and economic activities. Ten percent of the world's shorelines are fronted by barrier chains[6] and half of these coasts consist of mixed-energy barriers backed by marsh and tidal channels and separated by numerous tidal inlets (Hayes 1979). The inlets provide a means of the tidal exchange between the coastal ocean and the backbarrier basin. Future coastal impacts to barrier coasts will largely be driven by SLR, but the extent of change will also depend on how barrier-related geomorphic features evolve in response to these forcings[7]. Although anecdotal examples and conceptual models have been put forward to predict the future of these systems, a rigorous quantification is needed to lower uncertainty and demonstrate

theorized feedbacks. To date, research using modeling to predict the effects of SLR on basin geometry and inlet sediment transport trends is divided. Some studies predict that tidal inlets will import sediment[8-13], which is supported by field evidence[14]. Alternatively, other studies suggest that sediment will be exported as sea level rises[15,16], retarding the ability of marshes to build vertically[15]. Here, we examine how gradual submergence of the backbarrier system by SLR will change the backbarrier morphology and hydrodynamics, which alters sediment transport trends for different grain sizes.

How sediment is re-distributed during the evolution of coastal systems depends greatly on the underlying hydrodynamics of the basin, which in turn are strongly influenced by the distribution of saltmarsh and tidal channels. Tidal asymmetry, the difference in magnitude or duration between ebb and flood currents, is produced by distortion of the tidal wave as it propagates through tidal basins[17,18]. This occurs when the mean water depth is small, such that the geometry of tidal channels and resulting flow patterns are significantly different between high and low tide[19]. Imbalance in the flood and ebb periods impacts the associated current velocities; a shorter period transporting the same volume of water generates a higher velocity. The nonlinear dependence of the transport of coarse sediments on

[1]Moffatt and Nichol, 601 Poydras St, Suite 1860, New Orleans, LA 70130, USA. [2]Boston University, Department of Earth and Environment, 685 Commonwealth Avenue, Boston, MA 02215, USA. [3]The Water Institute, 2021 Lakeshore Dr., Suite 310, New Orleans, LA 70122, USA. ✉e-mail: igeorgiou@thewaterinstitute.org

velocity means that even a slight asymmetry in velocity produces net bedload (sand) transport, facilitating sediment import if tides are flood-dominant or export if tides are ebb-dominant[7,19]. However, for suspended load (clay/fine silt), slack water duration dictates residual flux rather than maximum current asymmetries, due to the slower rate of settling[19–21]. Residual flood-directed transport of fine-grained sediment is enhanced when channel depth decreases in a landward direction, or if the velocity variation is slower near the pre-ebb slack than the pre-flood slack. This trend in net sediment transport can be counteracted by wind waves, which, due to the shallower water depths, maintain more sediment in suspension near pre-flood slack than pre-ebb slack, when tidal flats are flooded[19,21]. Under these conditions, variations in the wetted area impact slack water durations and therefore, the net transport of fine sediment. The direction of the net transport is controlled by the elevation of intertidal regions, with high intertidal regions compared to the channel depth being characterized by longer pre-ebb slack, promoting fine-grained import, while a greater area of lower intertidal regions has the opposite effect, with longer low water slacks and net fine-grained export. A large intertidal area also leads to slower drainage from flats to the channels during the ebb; this water surface gradient leads to a peak in ebb-velocities that can result in residual ebb-directed transport of coarse-grained sediment[19,21,22].

If changes in basin morphology are neglected, then rising sea level will gradually increase water depths in the backbarrier throughout the tidal cycle. While the change in depth will be small relative to deep channels, water depths over the marsh platform could increase substantially. The increased depth over intertidal areas at high tide decreases frictional effects and allows faster propagation of the flooding tidal wave crest within the basin[22]. Relatively small deepening of the channels due to SLR will have little impact on the friction of ebbing flows within the channels. Finally, increasing tidal prism will also enlarge the channel cross-section, thus, with faster propagation of the flooding tide, there will be a smaller discrepancy between flooding and ebbing durations such that ebb-dominance is reduced. Thus, the gradual loss of intertidal storage area due to marsh submergence coupled with marsh edge erosion will serve to re-establish the natural flood-dominance of a progressive wave (due to nonlinear effects) or co-oscillating wave in moderate-length basins (due to frictional-effects)[19,22]. Because sediment transport is related to a power function of velocity[22], a slight shift to stronger flood versus ebb velocities will lead to net landward sediment movement through the inlet and within backbarrier channels[22]. For example, field evidence of landward sand transport in flood-dominated backbarrier by channels has been documented in Willipa Bay, Washington[23] and in Essex Bay, Massachusetts[24].

The response of tidal inlet and basin systems to SLR has previously been investigated with both semi-empirical or rule-based, and process-based models[25]. Several researchers have applied a sediment equilibrium model, aggregating morphologic response into volume changes of tidal flats, inlet channel, ebb-delta, and adjacent coast elements[9,26]. At two inlets along the Dutch Wadden Sea, the model predicts that increasing SLR rates induces a disequilibrium in element volumes that is compensated by expanding the inlet cross-section, as well as erosion of interior flats and the ebb-tidal delta[8]. Recent studies have also suggested that the demand of sediment in tidal basins is a product of increasing accommodation related to SLR[12,13]. Additional research has focused on process-based, morphologic models of conceptual tidal basins with varying geometries[27,28]. Using a schematized model of the Ameland Inlet[9], one study found that initially flood-dominant hydrodynamics are enhanced with SLR, resulting in increased import of sand that is partially supplied from the eroding ebb-delta. Despite consistent sediment import, the tidal channel network and tidal flats are only maintained under low rates of SLR (0.2 m increase from 1990 water level by 2100). Resulting morphologies show the development

of relatively deep, narrow backbarrier tidal channels with multiple ebb-dominant channels dissecting the ebb-tidal delta[9,29]. This work was expanded to incorporate the development of a laterally unconstrained inlet-basin system with different SLR rates and varying tidal ranges[30,31]. Results show that, for a SLR rate of 5.6 mm/y, tidal prism increases due to a lateral expansion of inundated area and consistent growth of ebb-delta shoals. However, trends in residual sediment transport direction (import or export) were found to vary with tidal range[30,31]. Long-term (millennial scale) modeling of a conceptual, large, elongated estuary (80-km long, 2.5-km wide) focused on tidal asymmetries responding to the loss of intertidal areas due to an imposed SLR of up to 6.7 mm/y[10]. In all SLR scenarios, the basin shifted from exporting to importing sediment, consistent with developing overtides due to tidal asymmetry[22,32], although at rates insufficient to prevent the intertidal area loss[10]. Recent work modeled SLR effects on sediment budgets in Plum Island Sound, Massachusetts, finding that SLR enhanced marsh sedimentation, while at the same time increasing the ebb-dominance of the system and export of sediment[15]. Another study tested the impacts of imposed marsh loss on six tidal basins along the U.S. East Coast, finding positive feedback whereby marsh loss reduced the sediment trapping efficiency of remaining marshes and enhanced ebb-dominance[16]. In a later investigation of Jamaica Bay, New York, the same authors[33] suggest that sedimentation inside the basin is derived from marsh edge erosion, as well as from marine sources. Finally, studies focusing solely on sedimentation of the marsh surface indicate conflicting projections. For example, a meta-analysis of existing data suggests that most marshes will be stable[34] which contrasts with eco-geomorphic modeling[35] indicating that marshes are highly vulnerable to SLR. A recent global meta-analysis shows that while marshes appear to be presently resilient, they are increasingly vulnerable with accelerating SLR due to auto-compaction and subsidence[36].

To address the very different conclusions of previous studies and investigate longer timescales and more alternative SLR conditions than possible with field studies, we undertake a rigorous, numerical investigation into changing basin morphology and hydrodynamics with SLR (Fig. 1). We improve upon previous research by using an erodible bed with a range of sediment classes (fine sand [200 μm], coarse silt [64 μm], clay [20 μm]) and study the response caused by SLR and a coupled SLR with a stipulated rate of marsh platform accretion. Thus, we allow for the full morphological evolution of the basin without intervention during the entire simulation period. We quantify relationships between SLR and the hydrodynamic and morphologic response of a representative tidal inlet-basin system. We specify a moderately sized basin geometry (12-km long) containing marsh and tidal channels, characteristic of mixed-energy barrier shorelines[37] that are common throughout the world (Table 1). Importantly, through numerical modeling, we provide a quantitative relation between predicted morphological evolution (based on empirical, equilibrium relationships) and expected hydrodynamic behavior (developed from numerical studies), with emphasis on the varying responses of both coarse and fine-grained sediment. By imposing SLR on the conceptual inlet/basin system examined here, we test the theory that an expanding inlet reduces friction and alters the propagating tidal wave, increasing accommodation space as well as shifting currents from ebb- to flood-dominance.

## Results and Discussion
### Basin Morphologic Response to SLR
Following a 200-year morphologic simulation, cumulative erosion and deposition for all scenarios: control with no SLR (Figs. 2a), 5 mm/y SLR (Fig. 2b), and 8 mm/y SLR with 3 mm/y marsh accretion (Fig. 2c), show a deepening of the inlet and backbarrier channels. Deposition occurs on backbarrier tidal flats and at the margin of the expanding ebb-tidal delta. Channel and tidal inlet deepening in the control simulation is a product of gradual adjustment of the backbarrier to the 1.5 m

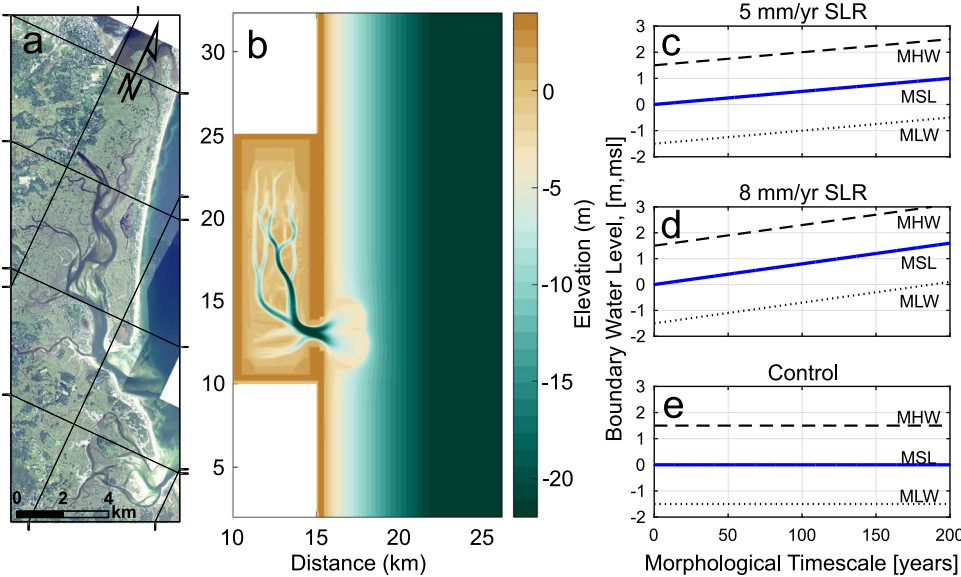

**Fig. 1 | Conceptual inlet-basin system and sea-level rise scenarios. a** Aerial photograph mosaic of Plum Island Sound (PIS) in northern Massachusetts, USA. **b** Conceptual inlet-basin system and initial bathymetry within the Delft3D hydrodynamic, sediment transport, and morphologic model domain. Initial bathymetry represents an approximate equilibrium condition reached after a 1.5 m amplitude semidiurnal, sinusoidal tide was imposed for several years of simulation time with a morphologic acceleration factor of 100. **c** Water level boundary condition for 5 mm/yr SLR case, consisting of a 1.5 m amplitude semidiurnal, sinusoidal tide superimposed with a linear rate of seal level rise, (**d**) Water level boundary condition for 8 mm/yr SLR case and (**e**) water level boundary condition for control case, consisting of same tidal conditions with no SLR.

amplitude tidal range. Greater depths are attained in the simulations with SLR (Fig. 2b, Fig. 2c) due to continuing basinal flooding causing a steady increase in tidal prism. Moreover, the simulation with 5 mm/y SLR shows higher low- and high-tide levels, which decreases the intertidal area by approximately 13%. The morphological trends of the basin/inlet system (Fig. 2, Fig. 3. Supplementary Fig. 1) are consistent with tidal prism equilibrium relationships of inlet cross-sectional area[38,39] and ebb-tidal delta volume[40].

Broad comparisons of aggregate changes in basin geomorphology are valuable metrics of system response to SLR[41,42] and were evaluated for each of the simulations (Fig. 3) to test the functional relationships among tidal prism and inlet area, ebb-delta volume, tidal channel volume, and tidal flat volume, because they are useful predictors of response to large scale disturbances including SLR[26,41,43,44].

**Table 1 | Mixed-energy barrier island-tidal inlet chains**

| Barrier System | Length of the Tidal Basin (Inlet to drainage divide, km) | Spring Tidal Range (m) |
|---|---|---|
| Merrimack Embayment, MA | 8–12 | 3.3 |
| Western Long Island, NY | 7–17 | 2.0 |
| South New Jersey | 4–9 | 1.9 |
| Northern Virginia | 4–10 | 1.6 |
| Southwest North Carolina | 3–11 | 1.9 |
| Northern North Carolina | 3–7 | 2.1 |
| Southern South Carolina | 3–14 | 2.3 |
| Georgia | 7–15 | 2.5 |
| Northern Florida | 6–13 | 1.9 |
| West Frisian Islands | 10–20 | 2.4 |
| East Frisian Islands | 4–11 | 3.1 |
| Denmark | 11–22 | 2.3 |
| Algarve, Portugal | 4–5 | 3.1 |
| Copper River Delta, AK | 6–12 | 4.0 |

With SLR, increasing tidal prism (Fig. 3a) is associated with increasing tidal inlet area (Fig. 3b), increasing volume of sediment sequestered in the ebb-tidal delta (Fig. 3c) and transfer of sediments from incising channels to accreting tidal flats (Fig. 3d). Results conform to established equilibrium relationships[39,40,45,46] and the transgression hypothesis[7] whereby the increasing tidal prism produces an expanding inlet cross section, ebb-tidal delta, and channel system, as well as a decrease in volume of tidal flats[8]. Without external sediment input, including sand that would be added from adjacent barrier islands, the inlet and bay channel incision provides sediment for ebb-delta and tidal flat accretion. For the control case, the tidal prism, channel volume, and tidal flat volume remain constant through time, albeit with small increases in inlet area and ebb-delta volume likely due to the initial basin configuration not yet being in full morphodynamic equilibrium since adaptation timescales of tidal basins to external conditions could be on the order of millennia[47]. Comparisons of predicted morphologic parameters using empirical functions of tidal prism[44,45] show that for the simulations with SLR, the inlet throat is larger than expected, while the total volume of tidal channels initially falls within predicted ranges (Supplementary Fig. 1). The larger inlet cross-sectional area is consistent with the overprediction of inlet incision, potentially due to the absence of wave-driven nearshore inlet sediment supply. However, the volume of the backbarrier channels, where nearshore processes have less influence, indicate an overall basin morphology that is consistent with natural mixed-energy systems[45].

## Hydrodynamic and Sediment Transport Response to SLR

The morphology of a basin-inlet system is continuously shaped by the evolving hydrodynamics, and thus, we compare peak flood and ebb velocities at the inlet for each tidal cycle as a general indication of the net sediment transport patterns. Since the transport of non-cohesive sediment is a higher-order function of velocity[48], an evaluation of peak velocities (in Fig. 4a) can be used to determine net direction and magnitude of sediment transport at the inlet (Fig. 4b). Our modeling indicates that peak ebb (channel-averaged) velocity exceeds the peak flood velocity until ~year 90 when they become equal for the 5 mm/y case. For the remaining 110 years of simulation, peak flood currents

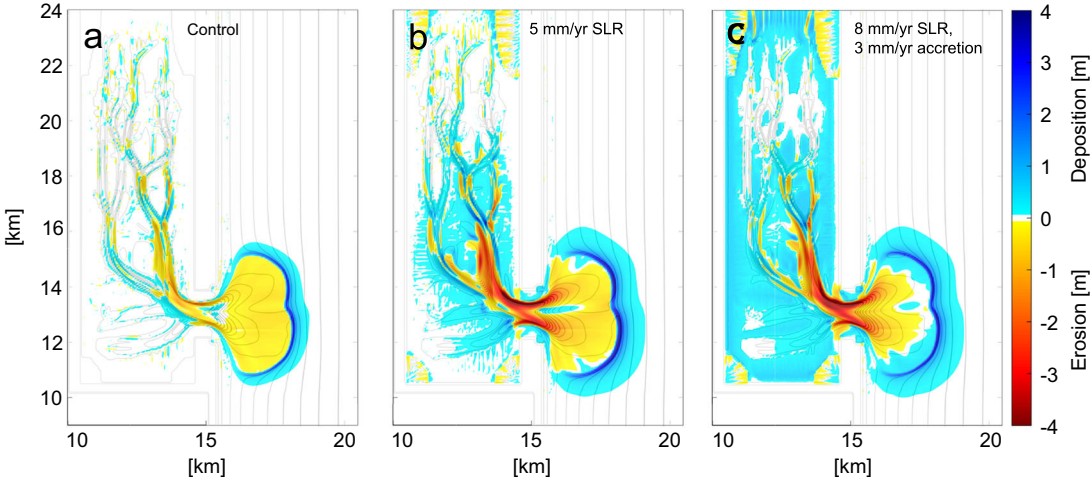

**Fig. 2 | Cumulative erosion and sedimentation from each scenario.** Cumulative erosion (negative) and sedimentation deposition (positive) for (**a**) the control (no SLR), (**b**) 5 mm/yr SLR cases, and (**c**) 8 mm/yr with 3 mm/yr accretion after 200 years of scaled morphologic simulation.

become dominant, reaching a maximum in year ~135 due to both an increase in peak flood velocities and diminished peak ebb velocities. Sediment transport trends mimic those of velocity, as expected (Fig. 4b), indicating for the first 90 years the inlet exports sediment and after that, it strongly imports sediment. In the control case, peak currents are relatively constant, and as such ebb-directed peak flow and sediment transport remain dominant for the entire simulation. For the 8 mm/y SLR case with manually imposed 3 mm/y marsh accretion (Fig. 4d), the shift to sediment import is modulated by the additional backbarrier sediment supply so that the fine sand and coarse silt fractions are imported for the years between 90 and 130.

For the control case, the constant, positive cumulative transport rate (Fig. 4c) indicates continued sediment export for all sediment fractions and total sediment load. For the 5 mm/y SLR case, the fine sand and coarse silt fractions are exported for the first ~90 years (87 and 93 years, respectively) after which time these coarser fractions are first imported and then strongly imported by year 135 (Fig. 4c). The model also shows that the total sediment flux discharging from the inlet persisted for another ~10 years after the net export of fine sand and coarse silt have ceased. Thereafter, the total sediment flux at the inlet shifts to importing sediment. Interestingly, fine sediment, such as clay, is continuously exported for the duration of the simulation. Due to its grain size, unconsolidated clay is easily suspended into the water column and can potentially be transported onto the marsh surface.

The shift from net sediment export to sediment import through the inlet for the 5 mm/y SLR case is also observed in backbarrier tidal creeks, where residual sediment transport patterns are ebb-dominant at the beginning of the simulation (Fig. 5a) and flood-dominant at the end (year 200; Fig. 5b). Before substantial SLR has occurred, residual transport in the main tidal channels is directed towards the inlet (Fig. 5a), but at the end of the simulation with 1 m of cumulative SLR, residual transport is directed landward in the main tidal channels. This trend indicates that sediment entering the inlet, due to its flood dominance, will be transported farther landward into the basin. Backbarrier residual transport patterns for the 8 mm/y SLR 3 mm/y accretion case are also consistent with the basin sediment import and export trends seen in the 5 mm/y SLR case; however, channel residual transport is ebb dominant during the full simulation except for during a period between years 90 and 130 when creeks are flood dominant. The eventual shift in net transport direction for both sand and total sediment is indicative of a shift in the tidal asymmetry from ebb- to flood-dominance, while slack-tide asymmetry (Fig. 6c) continues to favor fine sediment export[19].

## Hydrodynamic predictions compared with theory

Our conceptual tidal basin is moderate in length, such that the water surface slope within the basin is nearly horizontal throughout the tidal cycle. Horizontal tidal velocity is 90 degrees out of phase with the water level fluctuations so that slack tides occur approximately simultaneously with the high and low tides. This lack of progressive wave enables a simplified framework for examining tidal asymmetry based on classical hydrodynamic studies[19] (Supplementary Fig. 5). For all cases, the basin is initially ebb-dominant, with this ebb-dominance persisting in the control case for the full simulation. The basin is characterized by deep channels and relatively high tidal flats. The initial morphology (referred to as Type I by Dronkers,1986) produces a basin spatial-mean depth that is greater at low tide than at high tide, favoring ebb-dominance with higher ebb-directed tidal currents[19,49,50]. Likewise, deep channels compared to the tidal amplitude and large high-tide storage also produce ebb dominance[22]. The initial modeled mean basin water depth (Fig. 3e) is much greater at low tide than at high tide due to the deep channels and high elevation of tidal flats. With continued SLR and without adequate sedimentation on tidal flats (even with a proxy for organic marsh accretion in the 8 mm/yr SLR, 3 mm/yr accretion case), mean depths at high tide increase while mean depths at low tide decrease, reducing the mean depth differential that favors ebb-dominance.

In Type I basins, the tidal wave propagates faster in the deep channels than on the shallow flats, leading to a strong current during the last stage of the ebb-cycle when slower intertidal drainage produces higher water surface gradients[19,51]. This process is visible in the relationship between stage and velocity in a backbarrier creek (see Fig. 6a). Peak ebb-directed currents occur near the average tidal flat elevation (representative of the marsh edge in many backbarrier systems) of +0.5 m throughout the simulation, when tidal flats are nearly fully drained after high tide. Peak flood currents, however, occur with higher stages when flats are already inundated. While extensive intertidal area remains even in later years of the simulation, increased water depth over the zone due to SLR reduces frictional effects that slow the propagation of high water, thereby reducing ebb-dominance[18,22]. The continued inlet incision similarly reduces ebb-currents, as the ebb-directed residual current (Stokes return flow) generated by the Type I geometry are applied over an expanding inlet area[50].

For moderate-length tidal basins without significant frictional effects, the acceleration around slack tide is proportional to the basin area, and inversely proportional to the rate of change of channel cross-sectional area[19]. For a Type I basin (deep channels and high, extensive

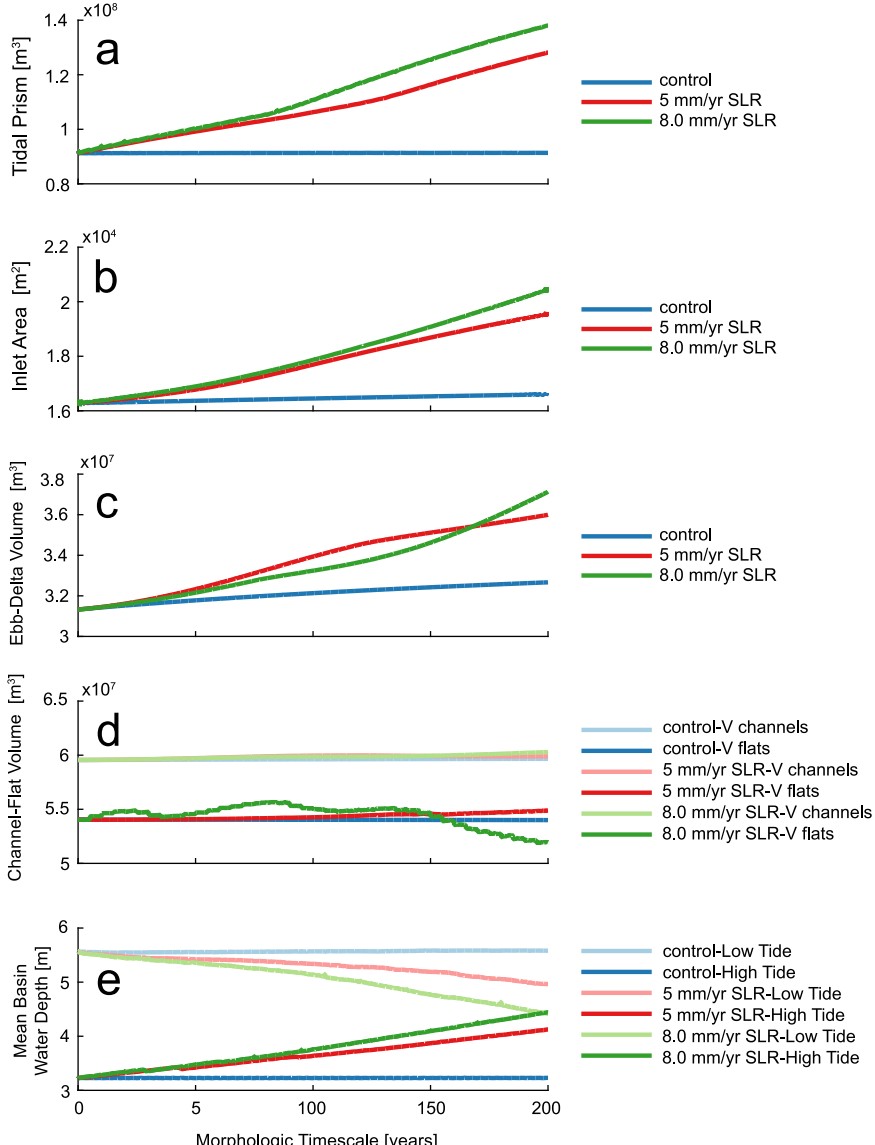

**Fig. 3 | Quantitative basin-inlet evolution parameters through time. a** Tidal Prism (volume of water passing through the inlet throat in each semi-diurnal tidal cycle), **b** Cross-sectional Area of inlet throat below the tidally-filtered water surface elevation, **c** Volume of sediment sequestered in the ebb-tidal delta, calculated as the positive volume between the delta bathymetry surface and a surface created by projecting an "undisturbed" cross-shore profile some distance away from the inlet (see Dissanayake et al 2011). **d** Volume of tidal flats and channels through time. Volume of channels is computed as the volume of water between the evolving tidal basin bathymetric surface and the initial Mean Low Water datum at the inlet. Volume of tidal flats is computed as the volume of sediment within the basin lying between the initial Mean Low Water and Mean High Water datums (see Dissanayake et al., 2011), and (**e**) Mean water depth in basin at high tide and low tide for the control (no SLR), 5 mm/yr SLR, and 8 mm/yr SLR with 3 mm/yr accretion cases.

Initially, all simulations have a much greater mean depth at low tide than high tide due to the deep channels and high elevation tidal flats. With continued SLR, without adequate sedimentation on tidal flats, mean depths at high tide increase with greater inundation over the marsh platform while mean depths at low tide decrease as expanding low tide inundation limits submerge lower elevation banks and tidal flats (**e**). SLR causes an increasing tidal prism (**a**), which produces an enlargement of the inlet cross-sectional area (**b**) and a growth in volume of the ebb-tidal delta (**c**). Likewise, there is a transfer of sediment from incising channels to accreting tidal flats (**d**). For the control case, tidal prism, channel volume, and tidal flat volume remain constant through time. Slight increases in inlet area and ebb-delta volume in the control case indicate that the initial basin configuration may not be fully in morphodynamic equilibrium with the forcing tidal conditions.

flats), currents accelerate and decelerate more quickly at high tide than at low tide. This condition favors an extended pre-flood slack water and fine sediment export[19]. While the increasing inundation depth over the extensive tidal flats with SLR shifts the basin away from the idealized Type I (favoring import of fine sediments), a counteracting effect reduces this effect. The tendency of sediment import increases as the period of high slack water increases, which allows more fine sediment to settle out of the water column over tidal flats[20]. However, as SLR increases, the water over the tidal flats is continually deepening, despite marsh accretion, which increases the required settling time for

particles to reach the substrate, thereby decreasing the amount of sediment deposited on the flats.

## Changes in flood/ebb asymmetry and resulting net sediment transport

The duration of slack tide before flood for both the control and the SRL case (Fig. 6c) remains longer than the slack tide before ebb throughout the simulation. The longer period of low water slack allows greater deposition of suspended sediment in interior tidal creeks while tidal flats are exposed. Persisting throughout the

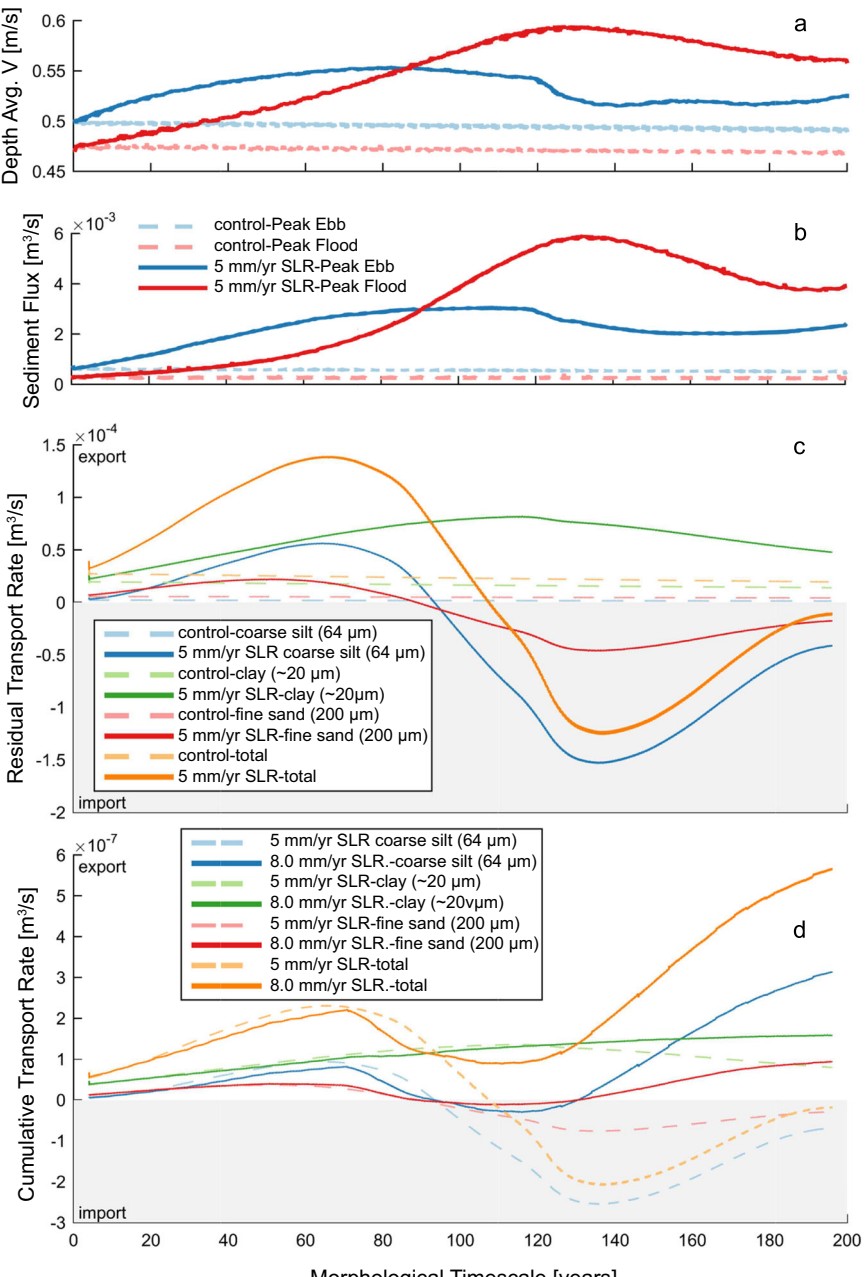

**Fig. 4 | Current velocities and sediment transport trends through time. a** Depth-averaged peak ebb and flood current velocities and (**b**) Sediment flux. Plot (**a**) indicates that the peak ebb-directed channel-averaged velocity through the inlet is initially greater than the peak flood- directed velocity. During the first ~ 90 years of SLR, both ebb-directed and flood-directed velocities increase, with flood peak currents exceeding ebb peaks after ~ year 90. In the control case, peak currents are relatively constant, so that ebb-directed peaks remain greater. Since the transport of non-cohesive sediment is a higher-order function of velocity[48] the trends in peak velocities in plot (**a**) are magnified in the plot of peak ebb- and flood-directed sediment fluxes (**b**). The shift to higher magnitude flood-directed flux with SLR indicates a shift from net sediment export to import. **c** Residual sediment transport (total and for individual sediment fractions) rates through the inlet for the control (no SLR) and 5 mm/y SLR cases. Positive values indicate sediment export while

negative values indicate sediment import. The SLR case is represented by solid lines while the control case uses lighter, dashed lines of the same color. The cumulative transport time series has been filtered to remove tidal fluctuations so that the plotted rates reflect residual transport. Note that the clay is exported over the duration of the simulation, however, coarse silt and fine sand shift to being imported by ~ year 90. **d** Residual sediment transport (total and for individual sediment fractions) rates through the inlet as in panel **c** comparing the 5 mm/y SLR and 8 mm/yr SLR with 3 mm/y accretion cases. Note the different y-axis scale from panel **c**. For the 8 mm/y SLR case with manually imposed 3 mm/y marsh accretion, the shift to sediment import is modulated by the additional backbarrier sediment supply so that the fine sand and coarse silt fractions are imported for the years between 90 and 130.

simulation, this discrepancy in slack tide durations promotes the continued export of fine sediments seen in all cases (Fig. 4c, Fig. 4d). The long-term export of clay in both SLR cases is an important finding, indicating an enhanced vulnerability of backbarrier salt marshes, dependent on the supply of fine suspended sediment for

mineral accretion[52–57]. The presence of wind-generated waves within the basin (a process not included in this model formulation) would naturally serve to further increase fine sediment export by reducing deposition over shallow tidal flats during the slack water period before ebb[19], and offshore waves would create a more-realistic ebb-

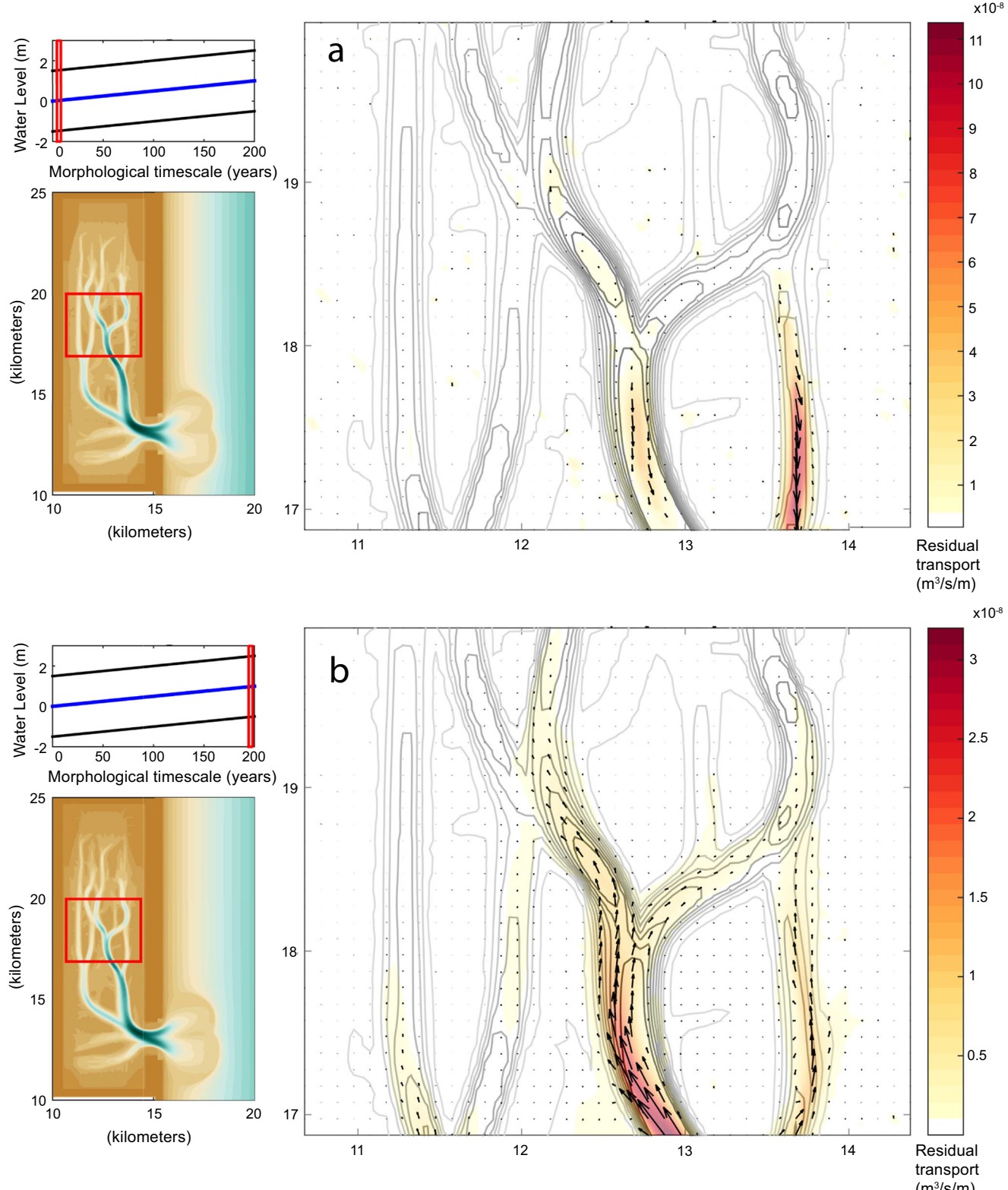

**Fig. 5 | Residual sediment transport trends.** Residual sediment transport patterns in the backbarrier channels at (**a**) the beginning and (**b**) at the end of the 5 mm/y SLR simulation. Residual was computed as an average during 28 semi-diurnal tidal cycles. Note the dramatic shift in net sediment transport from export to import at 0 and 200 years.

delta morphology characterized by an arc-shaped bar and single main channel.

The impacts of vegetation on the flow across the marsh surface and resulting influence on sediment transport patterns are also neglected in the current model, though results from previous work aid in determining the implications of this simplification. Vegetation would colonize tidal flats once the elevation exceeds mean sea level[58] (Fig. 2), which would enhance fine sediment deposition and vertical

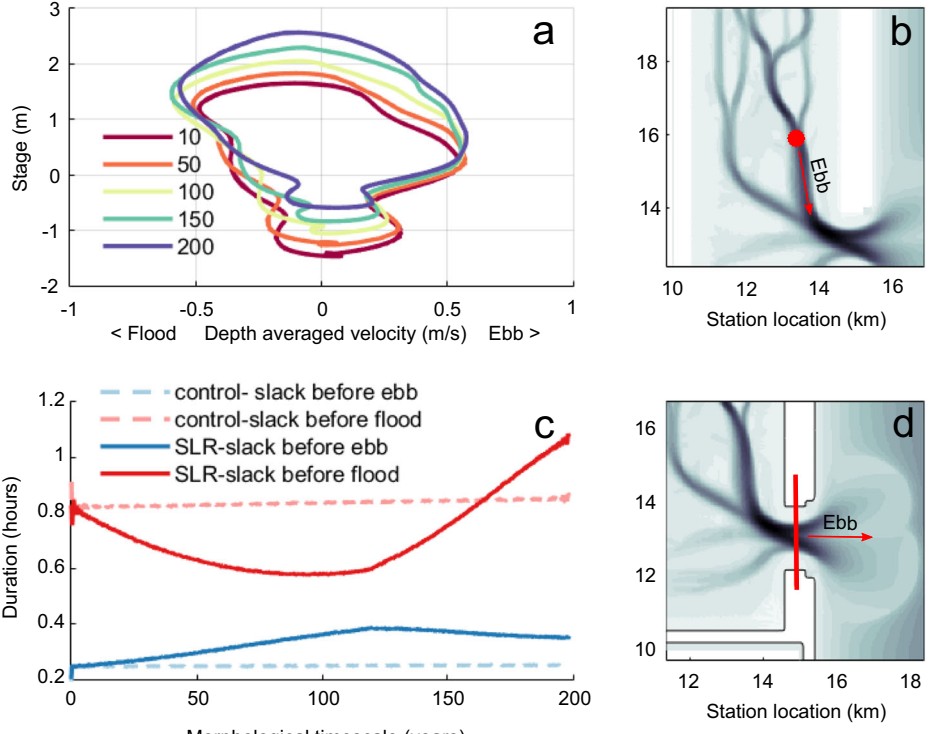

**Fig. 6 | Inlet and backbarrier hydrodynamics. a** Stage-Velocity curves through time at a point within a backbarrier tidal channel for the 5 mm/yr SLR simulation. The location of the point is indicated by the circle while the positive, ebb-tide direction is noted with the arrow in panel **b.** The curves are plotted for a single tidal cycle occurring near the start of the simulation (10 years) and in subsequent years until simulation ends at year 200. **c** Duration of slack tide before the ebb and flood tidal cycles at the tidal inlet through time. Durations were computed using the channel-averaged velocity along the inlet transect as plotted in panel **d**.

accretion[59,60]. Accretion of the marsh platform, as well as building subtidal flats to elevations conducive to vegetation colonization is highly dependent on the suspended sediment concentrations in the inundating tide[53,61–64]. The persistence of tidal distortion and the continued export of fine sediment would gradually reduce the supply available for deposition within the basin, including vegetated regions[57,65,66]. This condition, if combined with acceleration in SLR would lead to an increased rate of inundation, an effect corroborated by a modeling study[67] of a tide-dominated, funnel-shaped estuary with eroding salt marshes, despite the presence of vegetation on tidal flats decreasing the export of fine sediments.

Elevated tidal flat regions relative to the rising mean high-water level, however, could maintain the Type I basin behavior due to deepening channels without increased tidal flat inundation. Vegetation within the basin would likely enhance channel erosion with SLR[68,69] such that the ebb-tidal delta volume would grow commensurate with increasing tidal prism. The eco-geomorphic processes that determine the rate of marsh and tidal flat accretion relative to SLR rates are generally site specific[34,56] and depend on presence of absence of vegetation as well as dominant vegetation species coverage[70] but could promote tidal flat submergence or persistence in the modeled conceptual basin, with corresponding impacts to the evolution of tidal asymmetries and ebb-/flood-dominance[70].

Overall, our results show a shift from sediment export to a period of import with SLR. This is counter to recent findings[16] indicating a positive-feedback whereby marsh loss reduces the sediment trapping efficiency of the remaining marsh and enhances ebb-dominance. This conclusion[16] was based solely on a reduction in backbarrier deposition of suspended clay, which was determined by manually (in their model) releasing a fixed quantity of sediment in the backbarrier and evaluating where it was transported and deposited after a period of 30 days without allowing the bed to erode or aggrade. Moreover, the study[16] accommodates increasing tidal prism by widening the channel without deepening the inlet, a process shown to reduce ebb-dominance (in our study). Additionally, our study shows that SLR deepens the inlet including the backbarrier channels, which enhances flood dominance causing an influx of coarse sediment. Our model demonstrates the importance of co-evolution of the inlet-basin system forced by SLR and increasing tidal prism. Furthermore, our study is consistent with previous findings showing that with increasing tidal prism: (1) fine-grained sediment is exported[15] (2) inlet cross-sectional area increases[39,46], (3) ebb-tidal delta expands[17], (4) backbarrier tidal creeks enlarge[33], and (5) sand is imported[9,31].

## Implications for coastal systems evolution

Our conceptual backbarrier and tidal inlet system is modeled after mixed-energy barrier island chains that occur throughout the world[37]. These systems are characterized by moderate-length basins (Table 1) with relatively deep channels supporting a mostly standing tidal wave signature[18,19,71]. Although we fashioned the conceptual basin after mixed-energy, mesotidal systems, we acknowledge that the examples provided in Table 1 may contain different hypsometries, grain size variability, and tidal ranges, leading to different results. Still, the basin lengths, tidal ranges and deep channels would tend to produce standing tidal wave regimes leading to similar hydrodynamics and ebb-dominated inlets. Moreover, these backbarrier systems have

comparable ebb-tidal deltas and backbarriers consisting of marsh or well-developed tidal flats incised by tidal channels. In simulating the hydrodynamic and morphologic response to SLR of a conceptual tidal inlet-backbarrier basin-ebb-delta system, the conceptual model of runaway transgression[7] is corroborated. The process-based model reproduces a basin trajectory, evaluated using aggregate morphologic parameters, that positively correlates with tidal prism, simulating many of the elements of a runaway transgression barrier model[72]. With SLR, the basin hydrodynamics shift from ebb- to flood- dominance due to changing basin hypsometry, though the shift is modulated with high marsh accretion rates whereby ebb dominance is eventually restored. Accretion on tidal flats does not keep pace with SLR, thereby reducing the channel-flat variation in tidal wave propagation speed at high tide that promotes higher peak ebb-velocities. Reductions in intertidal area due to SLR create a negative-feedback that shifts the inlet from exporting to importing coarse sediment, while at the same time expanding the volume of the ebb-tidal delta. When the supply of sediment within the basin is greater due to an imposed marsh accretion, coarse sediment is imported for a period before the basin shifts back to exporting sediment. The coarse sediment fraction moving into the basin is due to the flood dominance (stronger maximum flood currents) at the inlet and within the major backbarrier channels, whereas the growth of the ebb-tidal delta is caused by the enlarging tidal prism. Sediment transported into the basin and onto the ebb-tidal delta will come, in part, from inlet deepening, but a primary source will come from the littoral system via longshore sediment transport[24,73].

Contrary to recent studies of mixed-energy tidal systems, particularly in the Netherlands[9], results from this study infer that the ebb-tidal delta will expand with SLR due to increasing tidal prism[17], which may reflect the use of a moderate-sized tidal inlet-basin system for the modeling. Sediment losses to the backbarrier system and trapped on the ebb-delta will lessen inlet sediment bypassing[7] and gradually deplete proximal sediment reservoirs, most likely from the adjacent barriers. This will make some barriers more susceptible to breaching and transgression.

We show that different sediment classes respond differently to SLR despite a hydrodynamic shift from ebb-to flood-dominant for periods of the simulation. The transport of coarse sediment switches from being exported to being imported for varying time periods, whereas fine sediment is continually exported. The long-term loss of fine-grained sediment to the coastal ocean further threatens backbarrier marshes whose sustainability relies on both increased biomass production and mineral sedimentation[59]. Further research is needed to evaluate the eventual disposition of the coarse sediment fraction entering through the inlet. For example, an observation that peat in Plum Island Sound marsh is composed chiefly of silt and fine sand[74] suggests that imported coarse sediment could contribute to marsh mineral accretion. Likewise, incorporating the effects of vegetation, and waves and major storms would aid in making the modeling predictions more robust.

## Methods

### Use of process-based Models
Hydrodynamics and sediment transport in coastal systems are becoming increasingly quantified, however, practical limitations on the spatial and temporal resolutions of field-surveyed data limit the degree to which process links can be established at longer time scales[75]. Physics-based numerical models allow for increasingly sophisticated representation of coastal hydrodynamics and morphology over timescales relevant to barrier island transgression and tidal basin submergence. This research provides opportunities to link process causation to long-term development[10]. Here we employ a broadly used numerical modeling system (Delft3D)[48,76] to simulate the evolution of a backbarrier/tidal basin system and study the processes driving sediment exchange and morphologic evolution therein.

### System Geometry and Physical Setting
We used an idealized basin so that relationships can be easily quantified and compared with theory. Both hydrodynamics and morphology are evolved over decadal to centennial timescales, complementing spatially- and temporally-limited field studies of natural systems. The model is forced at the open-ocean boundary with a tidal signal superimposed by varying SLR scenarios. The evolving inlet and basin hydrodynamics are tracked using time-series outputs of backbarrier water level and inlet throat tidal currents. Sediment volume change for the morphological elements is calculated using bathymetric surface differencing, and sediment exchange between the basin and open coast is tracked using integrated fluxes through the inlet.

The conceptual inlet-basin system we used for the model experiment was evolved dynamically with fully coupled hydrodynamics, sediment transport, and morphology from an initial flatbed bathymetry (Fig. 1b). The model hydrodynamic and morphologic grid consists of an elongated basin of approximately 15 km by 5 km (with 50 m resolution) and a 30 km alongshore by 10 km cross-shore section of the nearshore. We started with a theoretical maximum basin depth of 2 m without predefined channels. We then forced the model with a 1.5 m amplitude semidiurnal, sinusoidal tide at the seaward boundary and ran a 2-year simulation to dynamically evolve the bathymetry using computed sediment fluxes to a near equilibrium condition employing a morphodynamic upscaling factor of 100 (Delft3D MORFAC feature), generally consistent with or less than values used in other similar modeling studies[9,45]. Using a symmetrical, undistorted tidal signal at the offshore open boundaries ensures that any emerging asymmetries are entirely due to variations in the inlet and basin geometry with water level increases[19] (distortion of the tidal wave over the nearshore zone between the offshore boundaries and tidal inlet is negligible). The final bathymetry following the 2-year simulation was used as the initial bathymetry for the experiments we performed here. While a 200-y morphologic timescale simulation is inadequate to achieve full dynamic equilibrium with no further bathymetry changes[47](Fig. 2a), both the volume of backbarrier morphologic features (Fig. 3) and net sediment fluxes (Fig. 4c) have reached steady-state conditions (indicating dynamic equilibrium of aggregate morphology) onto which perturbations could be imposed. Basin dimensions are fixed, mimicking the anthropogenic infrastructure or steep uplands that limits lateral migration of many coastal wetlands[77]. The simulated tidal channels are generally deeper than those found in prototype tidal basins but are consistent with experimental channels generated using similar methods[29] (Supplementary Fig. 2, initial basin hypsometry at start of experiments).

### Basin Strategy
We purposely chose basin dimensions (5 km by 15 km) and tidal range (3 m) that are representative of backbarrier systems along mixed-energy barrier island coasts (sensu: Davis and Hayes 1984)[78], for example Plum Island Sound in northern Massachusetts (in Fig. 1a). As seen in Table 1, mixed-energy coasts with a range of physical settings (like our basin model) exist throughout the world; barrier island coasts comprise 10% of the world's shoreline. Essential characteristics of these backbarriers include their moderate basin lengths and relatively deep channels that in combination produce a Dronkers Type 1 standing-wave tidal signature. In addition, like our modeled system, their backbarriers consist of a marsh (or upper intertidal flat, e.g., East Frisian Islands, Copper River Delta barriers) incised by tidal channels and an accompanying tidal inlet fronted by well-formed sandy ebb-tidal deltas. Moreover, as demonstrated by examples in the Table 1, the vertical accretion rate chosen in the model is well within range of mixed-energy barrier systems. Finally, it should be noted that modeling results do not change by using a more symmetric drainage geometry because the hydrodynamics (yielding standing wave conditions) are dependent on basin length from the inlet mouth and channel morphology.

## Experimental Design

Multiple hydrodynamic and morphologic simulations were run for comparison: a SLR case where a 5 mm/yr linear SLR is superimposed on the 1.5 m amplitude semi-diurnal, sinusoidal tide, a control case with the same tidal boundary conditions without SLR, representing still-stand, early Holocene conditions[79] and a higher SLR case (8 mm/y) with marsh platform accretion of 3 mm/y manually imposed every 5 years of simulation. With the same rate of SLR relative to marsh accretion (i.e., 5 mm/y), the 8 mm/y SLR run investigates the influence of basin elevations and hypsometry interacting with higher SLR rates expected after 2050[1] and the influence of any organic accretion that is not included in other simulations. A 5-yr marsh platform accretion update interval was chosen to maximize the simulation time between manual bathymetry updates while limiting the cumulative accretion between updates to a reasonably small value that would not impact model stability when instantaneously imposed (sensitivity testing also showed minimal differences in cumulative flux results among update intervals of 2, 5, and 10 years). Additional simulations with varying rates of SLR and marsh accretion were conducted, with the three cases presented here being representative, and respective elevations of the Mean Low Water (MLW), Mean Sea Level (MSL), and Mean High Water (MHW) tidal datums during the three simulations shown (Fig. 1c, Fig. 1d, Fig. 1e). While previous modeling studies of inlet and basin response to SLR used a single, representative sand fraction[9,10,30], we employ three sediment classes: 200 μm (fine sand), 64 μm (non-cohesive coarse silt), and cohesive clay with a fall velocity of 0.25 mm/s (corresponding to an approximate median grain diameter of 20 μm assuming Stokes' settling;[48]). The use of multiple sediment classes provides the potential to capture differences in net behavior of fine and coarse sediment under dynamically evolving basin geometry and changing tidal asymmetry. We employed a homogeneous and uniform stratigraphy throughout the basin and modeled area based on equal volumes of the three grain sizes. The sediment transport module used a critical shear stress for the erosion of cohesive sediments of 0.25 Pa, vertically uniform mixed stratigraphy initially composed of approximately 43% fine sand, 43% coarse silt, and 14% clay, and horizontally uniform sediment properties. The erosion and deposition of cohesive sediment are calculated using the Partheniades-Krone method (critical shear stress of 0.25 Pa, and erosion rate parameter of 0.0001 Kg/m²/s), and transport of noncohesive sediment is calculated using the Van Rijn formula[48,80]. We specified a sediment thickness of approximately 7 m to avoid limiting erosion, beyond which we used a non-erosional depth of 7 m to represent reasonably consolidated material or bedrock that would not easily erode for the timescales simulated. The dry cell erosion factor was set to one to allow tidal channel lateral migration (Supplementary Table 2).

Inclusion of wind-waves in the model show their impact on sediment transport is limited and localized, particularly in the backbarrier due to the narrow basin geometry. The ebb-tidal delta and adjacent shoreline are areas where waves augment sediment transport processes. Our modeling results of variable conditions show that waves produce limited net sedimentological change in the basin. For example, using a scenario that simulates 100 years and imposing a 5 mm/yr SLR, identical to a previous run, and using a schematized wave with an oblique wave approach (75-degree), a 6-second period, and 0.25-m wave height, we see that the seaward portion of the ebb-tidal delta undergoes uniform erosion while the region inside the inlet experiences areas of slight erosion and deposition (see SI for more detailed results and discussion). It is noted that events triggering wave-induced sediment transport inside the basin are highly infrequent and exceedingly diminutive when compared to tidal transport. We acknowledge the importance of waves in modifying the ebb-tidal deltas and resuspending fine-grained sediment in certain nearshore settings[81], which affect basinal sediment exchange with the coastal ocean. However, this study focuses primarily on the backbarrier basin and sediment transport is dominated by tidal currents, particularly at the inlet entrance, main channel, and inside the basin.

## Implementing an accretion rate

Representative accretion rates were chosen based on measured values in both vegetated and unvegetated regions, according to elevation. Applying an elevation-related accretion rate, using empirical data, has previously been shown to provide very similar results to the implementation of a vegetation-based morphological model, such as the Marsh Elevation Model (MEM) that computes accretion rates based on biomass production and inorganic sedimentation, but with lower computational expense. For example, previous analysis[82], using an accretion rate based on average values from the literature, estimated the conversion of a New England marsh from high to low marsh by 2055 (Authors Fig. 5, panel 1)[82]. Using a high rate of sea level rise condition, similar to the RCP 4.5 scenario previously used[82] (both reaching +1 m in 2100), in the same marsh system, but using a fully parameterized MEM marsh accretion model[83] found the conversion to occur between 2045 and 2070, and most likely around 2058.

## Vegetation roughness sensitivity analysis

The impacts of vegetation on the flow across the marsh surface and resulting influence on sediment transport patterns are neglected in the model experiments presented herein. Additional sensitivity analyses were performed that utilize the Delft3D trachytope roughness schematization (as a function of vegetation stem height and density according to the *Baptist 2* model[84]) that increases bed roughness over the marsh platform[84], defined as the basin areas with elevations between the time-varying MSL and MHW elevations. For the 5 mm/y SLR case, inlet sediment fluxes were negligibly impacted despite minor increases in marsh deposition (Supplementary Fig. 3). For the 8 mm/y SLR, 3 mm/y marsh accretion case, enhanced bottom roughness slightly increases ebb currents during the middle portion of the simulation (approximately years 80 through 140) which increases export of coarse sediments relative to the simulation with no trachytope roughness, though flux trends outside of this period are minimally impacted (Supplementary Fig. 4).

## Evaluation of water and sediment fluxes and analysis

The analysis of net sediment transport trends was made possible by use of a cross-section across the inlet of the system, to correctly account for all sediment fluxes for each of the sediment classes used. Sediment fluxes at this location were integrated to determine cumulative transport. Similarly, inlet geometry information such as inlet area, tidal prism volume, instantaneous and cumulative water flux, water velocity, and water level information were also evaluated at this location to establish correlations and facilitate analysis used in the paper, including comparison to theory.

# Data availability

The data generated by the simulations in this study can be reproduced using the modeling setup files provided in a public repository (https://doi.org/10.5281/zenodo.7672511). All other data generated are provided in Supplementary Information/Source Data Files.

# Code availability

The numerical model used in the analysis is the Delft3D-4 modeling suite and is available in the public domain. The software and computer source code are available at https://oss.deltares.nl/web/delft3d/downloads.

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

## Acknowledgements

We thank Christopher J. Hein for discussions that improved this research, and the University of New Orleans, Office of Research and Sponsored Programs (grant CON00002365 to IYG) for support of KCH to conduct initial numerical modeling experiments and analysis.

## Author contributions

I.Y.G. and D.M.F. conceptualized the study, and I.Y.G. and K.C.H. developed the methodology. K.C.H. carried out numerical modeling experiments and developed visualizations with input from I.Y.G. and D.M.F. I.Y.G. supervised the study with input from D.M.F. and Z.J.H. K.C.H. and I.Y.G. wrote the original draft, and I.Y.G. and D.M.F. edited, revised, and re-wrote portions of the manuscript with input from K.C.H. and Z.J.H.

## Competing interests

The authors declare no competing interests.
