## [Peer Review File · Nature Communications]

Long-term sea level rise modeling of a basin-tidal inlet system reveals sediment sinksREVIEWER COMMENTS

Reviewer #1 (Remarks to the Author):

I thank the authors and the editor for the opportunity to review this manuscript. I thoroughly enjoyed reading the results and discussion and recommend the paper for publication pending minor edits.

Key results

The authors use idealized modeling based on realistic estuarine geometry to explore the effects of sea-level rise (and background intertidal accretion) on backbarrier and inlet sedimentation and hydrodynamics. The key important finding is that, as predicted in the runaway transgression conceptual model, the ebb tidal delta and flood tidal delta eventually sequester more sand, which along a real barrier coast would starve the adjacent barrier islands of sand (in addition to the sand scoured from the inlet thalweg). Additionally, the authors find that estuaries have the potential to import sediment in the long run, an idea that runs counter to other recent work. As such, the work contributes meaningfully to the debate around the fate of salt marshes (and the general response of mixed-energy backbarrier estuaries) in the face of sea-level rise.

Validity

The model-derived data and interpretations are robust. While exact detail and assumptions can be debated and discussed, the approach and conclusions are scientifically valid. The authors' conclusions are supported by the results of the modeling.

Significance

These results are broadly relevant and significant. The authors provide new insights on estuarine response to sea-level rise and add to ongoing conversations about the morphodynamic response of mixed-energy backbarrier marshes to sea-level rise. The results are relevant not only to active discourse about the fate of salt-marshes, but the big-picture takeaways and conclusions will aid coastal managers in understanding potential pathways along which estuaries (and their associated sediment budgets) may evolve with sea-level rise.

Data and methodology

I am familiar with the hydrodynamic and morphologic data the authors generated and am comfortable evaluating its validity. The authors rely on idealized physics-based numerical modeling to address a timescale that cannot be reached based solely on field data. The model used (Delft3D) is one of a few commonly used for this type of scenario modeling and has been around for nearly two decades. The authors clearly state their assumptions and the limitations of the approach.

Analytical approach

The authors primarily rely on straight-forward timeseries data and surface-change maps. These are more than sufficient at communicating the relevant results.

Suggested improvements

The authors intentionally focus on three key scenarios in their modeling and justify their reasons for that. Additional data from the three modeling scenarios or other scenarios run does NOT need to be presented here.

Clarity and context

I found all parts of the text clear and easy to follow. The authors rely on numerical methods and a physics-based model, which can be challenging to describe to all readers. However, the authors do an excellent job of clearly explaining their methods and assumptions. For the rare occasions where clarity could be improved (select figures, occasional lines of text), I direct the authors to my line item comments.

References

The authors cite a broad literature base, including numerically-focused papers, field-informed papers, and those that are a mix of the two approaches. The authors are clearly familiar with both the foundational literature on estuarine dynamics, as well as more recent literature relating those dynamics to the fate of salt marsh ecogeomorphic systems. I do not see any noticeable gaps in the literature cited. Furthermore, the authors center this work in the relevant literature and engage with that literature in the discussion.

Line Item Comments and Suggestions:

Main Text

Line 219 to 221 – This text reads like discussion. I suggest moving it there.

Figure 3 Panel E – The caption is doing a lot of the work here. It certainly helps clarify what is going on. I suggest a minor tweak to the y-axis label so that it reads “Mean Basin Water Depth (m)” to match what is described in the caption.

Figure 4 Panel B – Y-axis label is hard to read and confusing. I think it is referencing total and individual fluxes but the caption could address that (similar to how the authors' treat panel C). I suggest remaking the axis label.

Figure 4 Panel D – There is no caption and it needs to be clearer that the reason for having C and D is simply to show the 5 mm/yr versus 8 mm/yr cases. The change in y-axis scale also needs to be clearer. I found myself staring at this figure trying to make sense of it for much longer than necessary. A caption as good as the others in this paper would go a long way.

Supplementary Materials

“SRL” (2 places) should be SLR. Suggest defining sea-level rise (SLR) on first instance in supplementary materials since it is separate from the main text.

Reviewer #2 (Remarks to the Author):

The manuscript describes a study dealing with a very important problem: the impact of sea level rise on tidal inlet systems. The findings would potentially be very relevant for the sustainable management of coastal system. The study is based on process-based morphodynamic modelling of an idealized tidal basin. Compared to a similar previous study (Dissanayake et al., 2012) the model is extended by including multi-fraction sediment transport, viz. three fractions (sand, silt & clay) instead of a single sand fraction. However, instead of focusing on a single real-world tidal inlet system (Ameland Inlet in the Dutch Wadden Sea) by the former study the present manuscript claims that the findings are relevant for mixed-energy tidal inlet systems that occur throughout the world. I think that this is a too far-reaching claim, as the model simulations concern only one idealized tidal inlet. Thus, no variation in basin geometry, initial bathymetry, initial bottom composition, and forcing conditions is considered. It is well known that the response of a tidal basin to (changing) sea level rise, e.g. whether (accelerating) sea level rise causes extra import or extra export, very much depends on its morphology (Friedrichs et al., 1990). The modelling study as presented by the manuscript can at most be relevant for a certain group/category of tidal inlets in the world. However, it is not clear which tidal inlets the modelled case would represent, as some essential characteristics of the idealized basin are not presented. It would help if the hypsometry of the basin was presented. Moreover, the modelled case does not represent mixed-energy tidal inlets as waves are ignored in the model.

Another concern of mine is that no validation of the used model has been presented. Not any comparison of the model results after the spin-up simulation, which are used as initial conditions (bathymetry, and bottom composition?) for the simulations with sea level rise, with any real-world case is presented. Even though the idealized case cannot be compared with any specific case, at least some essential morphological and sedimentological characteristics should be qualitatively compared with real-world cases. One of the findings that the fine sediment fractions, especially the clay fraction sediment, are exported due to sea level rise is against the finding from a recent study for the Wadden Sea based on field observations (Colina Alonso et al., 2021). I am afraid that the finding in this modelling study is due to some specific setting concerning e.g. the bottom composition.

The conclusion that “Sediment sinks on the ebb-delta and within the backbarrier system will be at the expense of proximal sediment reservoirs” is not based on the presented modelling results. As waves are not included in the model the model results do not show sediment transport from the adjacent coasts to the ebb-tidal delta.

The manuscript does not include all essential information concerning the model set up. Which transport

(/erosion) formulations are used for the three sediment fractions? How do you distinguish intertidal flat and marsh plat form (or are they the same)? Unless the model files are made available the presented information does not allow someone else to reproduce the presented modelling results.

Details

Line 71-73. Also for fine sediment maximum current asymmetry has influence on the net residual flux in addition to the asymmetry in slack water duration. Which of the asymmetries is more important differs from case to case.

Line 79-82. The statement on the relation between slack water duration asymmetry and direction of net transport is wrong. The correct relation is: longer pre-ebb / HW slack promotes import of fine sediment, and vice versa.

Line 92-93: what is meant by “the flood tidal wave”? Highwater? Or rising tide?

Line 108-109. The predicted decreases of the tidal flat volume and ebb-tidal delta volume are not due to erosion but due to the rising sea level. The volume of intertidal flat in the basin is defined as the sediment volume between MOVING LW and HW. More recent results on impact of sea level rise on the Dutch Wadden Sea are reported by Lodder et al. (2022) and Huismans et al. (2022).

Line 210. What is “cumulative transport rate”? Cumulative transport often (e.g. in the output of Delft3D) refers to the time integral of sediment transport rate, but in Fig.4c it has the unit of sediment transport rate.

Line 292-293. The tidal flat above MSL vegetated? To my knowledge, the transition between bare mud flat and vegetated marsh is around MHW. I do understand that the used model only includes a single tidal constituent, so MHW cannot be used.

Line 327. 911?

Line 396. The open boundary? There are three boundaries in the sea.

Line 397. The tidal wave also deforms as it propagates in the shallow sea outside the inlet.

Line 456-458. Waves also influence residual sediment transport as they can cause sediment concentration difference between inside and outside the inlet.

References:

Colina Alonso, A., van Maren D.S., Elias E.P.L., Holthuijsen S.J., Z.B. Wang, 2021. The contribution of sand and mud to infilling of tidal basins in response to a closure dam. *Marine Geology* 439 (2021) 106544.

<https://doi.org/10.1016/j.margeo.2021.106544>.

Friedrichs, C.T., Aubrey, D.G., Speer, P.E., 1990. Impact of relative sea level rise on evolution of shallow estuaries. In Cheng R.T. (ed.), *Residual current and long-term transport*, Springer-Verlag, New York.

Huismans, Y., van der Spek, A., Lodder, Q., Zijlstra, R., Elias, E., Wang, Z.B., 2022. Development of intertidal flats in the Dutch Wadden Sea in response to a rising sea level: Spatial differentiation and sensitivity to the rate of sea level rise. *Ocean and Coastal Management* 216, 105969.

<https://doi.org/10.1016/j.ocecoaman.2021.105969>.

Lodder, Q., Huismans, Y., Elias, E., de Looff, H., Wang, Z.B., 2022. Future sediment exchange between the Dutch Wadden Sea and North Sea Coast - Insights based on ASMITA modelling. *Ocean and Coastal Management* 219, 106067. <https://doi.org/10.1016/j.ocecoaman.2022.106067>.

Reviewer #3 (Remarks to the Author):

General:

The manuscript is generally well written, and of high quality.

I really like to detail and thought put into the beginning of the discussion.

Unfortunately is the chosen main message not always clear which is reflected in inconsistencies in the manuscripts structure.

More specifically:

1. it is unclear whether to focus lies on the morphological backbarrier basin response to SLR or the interaction between vegetated and unvegetated platform in response to SLR
2. it is unclear whether the main question focuses on the impact of multiple grainsizes on the basin response and how applicable this would be to other systems?
3. how important is the expansion of the ebb-tidal data, how is this incorporated in the model?

I am not saying these sub-questions could not be linked, however in the current form this link remains unclear to the reader  additionally what is the main message the reader can take home from this paper, the current manuscript structure more closely reflects a submission to a specialized journal like JGR than nature communications?

4. Broader impact, the manuscript portrays the results to be generally applicable to backbarrier salt marshes, however it remains uncertain how representative the physical- and boundary conditions of the chosen test case (i.e. Plum Island sound) is for barriers world wide. For instance even along the US east coast we encounter a wide variety of back barrier basin sizes and forcing conditions (mixed, mixed-tide, wave-dominated) at the seaward side. For instance mixed energy systems as indicated in table.1 have vastly different dynamics in barrier inlet and thus sediment fluxes, for instance waddensea inlets act as sinks for littoral sediment transport except sediment transported by sediment bypassing around the ebb-tidal delta.

5. Focusing on the specific model case study, how well is the model capable reproducing the hydrodynamic characteristics of the model system, e.g. tidal asymmetry, slack water duration. How well is the model, on average able to reproduce the distribution of different sediment types as shown in Fig.1, finer (clay silt) and coarse (sand) sediment types?

6. Vegetation: I appreciate the sensitivity study carried out by the authors, testing the impact of vegetation presence on the platform. It is not fully clear which vegetation parameters have been tested, where on the platform they were implemented?

abstract:

The abstract is not easy to understand and fails to summarize the main points.

More specifically:

Line 21: from the abstract the effect of imposed salt marsh accretion is unclear, which trend gets diminished, the shift from export to import?

Line 22: What do the authors mean by the trend for sand is more subdued? How does clay import or export change with or without saltmarshes?

Line 23: "This loss of sand will threaten barrier resources and infrastructure" remains unclear

main text:

generally difficult to follow

line 33: rephrase "the ~37% of coastal population that live there"

line 37: what are the authors referring to, what type of coastal systems do they focus on?

is this true for all systems,.. rocky and sedimentary coasts? cliff erosion and dune overwash might appear at a specific type of coastal systems, however are they the main driver of coastal hazards ?

Line 40: are the barriers themselves heavily developed, which systems do the authors have in mind? or are the authors referring to the back-barrier basin?

Before this line, for the reader to appreciate the subject of this study a short introduction in barrier-systems needs to be given, what kind of morphological units can we distinguish, how can they structure barrier dynamics?

Line 45: What do field observations reveal ? Is this proposed gap a problem with our ability to model barrier systems or a trend also observed across real systems ?

The proposed summary is very difficult to follow:

"Some studies predict that tidal inlets will import sediment 8–10 or that tidal range will control net inlet transport patterns 11, and more recent studies suggest that sediment will be exported as sea level rises 12,13, retarding the ability of marshes to build vertically".

 how is this sentence linked to the underlying processes increasing vs. decreasing tidal prism, increasing vs. decreasing load of suspended or bed load sediment,.. are there also rivers discharging sediment into the backbarrier lagoon? what is the impact of waves, vs. tides, which process is dominant, when and where?

Line 49: the research question is unclear ?? in general backbarrier lagoons are characterized by finer sediment types, such as fine sand, silt and clay. Is fine sediment considered ? What do the authors mean by sequestration of sand?

Line 52: Especially flood tidal delta evolution is governed by waves by littoral drift (longshore transport) and sediment bypassing caused by oblique coastal waves !  please revise , .e.g. De Swart, H. E., and J. T. F. Zimmerman. "Morphodynamics of tidal inlet systems." Annual review of fluid mechanics 41.1 (2009): 203-229. what kind of barriers are the authors referring to, tide-, mixed- or wave-dominated barriers since the relative contribution of processes will vary as will ebb and flood-tidal delta volumes, shapes and dynamics? e.g. using the classification of Hayes 1979

Line 58/supplement: I recommend for the authors go over the supplementary text and remove present typos:

Moreover, the discussion on the marsh platform keeping pace with sea level rise, seems to jump between an oversimplified 0d approach(1st paragraph) and a more detailed/realistic marsh platform, levees and channel approach (2nd paragraph).

 i suggest for the authors to:

1st merge this to points-of-view in a cohesive text

2nd since salt marshes seem to exert important control on the backbarrier system in respect to the posed research questions I propose to move at least part of it to the main text !

Line62-85: is an excellent paragraph and in my opinion should be moved into the main focus of the current study

Line 98: Will the net import of sand into the inlet lead to different cross-sectional areas in tidal channels and shallower backbarrier basin depths ? Will sand be also transported onto the marsh platform? Is an increase in transport capacity to be expected leading to coarser particles being deposited on intertidal platforms. If so has this been observed in previous field studies? the provided references mainly point to a model study (friedrichs and aubry '88), and a review from Fitzgerald '08) ?

Line 122- 138: different impressions on marsh survival and ebb-flood dominance originate from field and model exercises with varying complexity? I think here the authors miss a chance to justify the current approach, i.e. what do we know from field studies what are their boundaries, what do we need models for (and what are their boundaries)

Line 138: current research question

Line 139: i think using a numerical model does not represent a theoretical investigation, or are the authors referring to something else?

Line 140-143: here the research gap and question remains unclear and the reader is provided with details, which are appreciated but unfortunately without sufficient context.

Line 143: "Thus we allow for full .." unclear compared to what kind of model?

Line 147: Table.1 incomplete how does sediment fluxes and tidal ranges compare?

Line 178ff: reference missing

Line 223-224: which figure shows slack-tide asymmetry ?

Line 229: rephrase

Line 239-288: is an excellent discussion which should be put more in the focus of this study

Line 253: it think the author mean infilling of the channels which could be stated more explicitly

Line 298: I am not sure the reference to Aubry friedrichs is applicable here

Discussion:

Line 313 if the a Main aim was to disproving results by Donatelli 2020, and thereby indicating a positive feedback between marsh loss and sediment trapping efficiency this should have been indicated earlier?

Line 398: was the bathymetry in some dynamic equilibrium? unclear using Fig.3 and Fig.4c?

Line 410: how comparable are the forcing compared to your system as in sediment supply and tidal range,... which might alter their dynamics as shown in line 244ff. Ebb-tidal dela volume is dependent on wave and tide dominance, which raises the question to how many system Plum Island is comparable too?

Line 419: see point Line 410

Line 446: specifics of the model setup is missing, sediment transport equations, treatment of mixed beds, if not in the main article this should be present in the appendix so the study stays reproducible.

Line 463: It is unclear how the MEM implemented, I understand that not the whole model can be described, however sufficient information should be present to interpret the results, how does accretion rate change over space?

Line 471: insufficient details describing vegetation roughness tests.

COMMENTS copied from the Assoc. Editor communication:

REVIEWER COMMENTS

REVIEWER #1 (Remarks to the Author):

I thank the authors and the editor for the opportunity to review this manuscript. I thoroughly enjoyed reading the results and discussion and recommend the paper for publication pending minor edits.

Key results

The authors use idealized modeling based on realistic estuarine geometry to explore the effects of sea-level rise (and background intertidal accretion) on backbarrier and inlet sedimentation and hydrodynamics. The key important finding is that, as predicted in the runaway transgression conceptual model, the ebb tidal delta and flood tidal delta eventually sequester more sand, which along a real barrier coast would starve the adjacent barrier islands of sand (in addition to the sand scoured from the inlet thalweg). Additionally, the authors find that estuaries have the potential to import sediment in the long run, an idea that runs counter to other recent work. As such, the work contributes meaningfully to the debate around the fate of salt marshes (and the general response of mixed-energy backbarrier estuaries) in the face of sea-level rise.

Validity

The model-derived data and interpretations are robust. While exact detail and assumptions can be debated and discussed, the approach and conclusions are scientifically valid. The authors' conclusions are supported by the results of the modeling.

Significance

These results are broadly relevant and significant. The authors provide new insights on estuarine response to sea-level rise and add to ongoing conversations about the morphodynamic response of mixed-energy backbarrier marshes to sea-level rise. The results are relevant not only to active discourse about the fate of salt-marshes, but the big-picture takeaways and conclusions will aid coastal managers in understanding potential pathways along which estuaries (and their associated sediment budgets) may evolve with sea-level rise.

Data and methodology

I am familiar with the hydrodynamic and morphologic data the authors generated and am comfortable evaluating its validity. The authors rely on idealized physics-based numerical modeling to address a timescale that cannot be reached based solely on field data. The model used (Delft3D) is one of a few commonly used for this type of scenario modeling and has been around for nearly two decades. The authors clearly state their assumptions and the limitations of the approach.

Analytical approach

The authors primarily rely on straight-forward timeseries data and surface-change maps. These are more than sufficient at communicating the relevant results.

Suggested improvements

The authors intentionally focus on three key scenarios in their modeling and justify their reasons for that. Additional data from the three modeling scenarios or other scenarios run does NOT need to be presented here.

Clarity and context

I found all parts of the text clear and easy to follow. The authors rely on numerical methods and a physics-based model, which can be challenging to describe to all readers. However, the authors do an excellent job of clearly explaining their methods and assumptions. For the rare occasions where clarity could be improved (select figures, occasional lines of text), I direct the authors to my line item comments.

References

The authors cite a broad literature base, including numerically-focused papers, field-informed papers, and those that are a mix of the two approaches. The authors are clearly familiar with both the foundational literature on estuarine dynamics, as well as more recent literature relating those dynamics to the fate of salt marsh ecogeomorphic systems. I do not see any noticeable gaps in the literature cited. Furthermore, the authors center this work in the relevant literature and engage with that literature in the discussion.

Line Item Comments and Suggestions:

Main Text

Line 219 to 221 – This text reads like discussion. I suggest moving it there.

Response: These lines have been moved to the Discussion

Figure 3 Panel E – The caption is doing a lot of the work here. It certainly helps clarify what is going on. I suggest a minor tweak to the y-axis label so that it reads “Mean Basin Water Depth (m)” to match what is described in the caption.

Response: We have changed the y-axis title to read: Mean Basin Water Depth.

Figure 4 Panel B – Y-axis label is hard to read and confusing. I think it is referencing total and individual fluxes but the caption could address that (similar to how the authors' treat panel C). I suggest remaking the axis label.

Response: The Y-axis has been changed.

Figure 4 Panel D – There is no caption, and it needs to be clearer that the reason for having C and D is simply to show the 5 mm/yr versus 8 mm/yr cases. The change in y-axis scale also needs to be clearer. I found myself staring at this figure trying to make sense of it for much longer than necessary. A caption as good as the others in this paper would go a long way.

Response: We fixed the caption for Panel D

Supplementary Materials

“SRL” (2 places) should be SLR. Suggest defining sea-level rise (SLR) on first instance in supplementary materials since it is separate from the main text.

Response: This has been done.

REVIEWER #2 (Remarks to the Author):

The manuscript describes a study dealing with a very important problem: the impact of sea level rise on tidal inlet systems. The findings would potentially be very relevant for the sustainable management of coastal system. The study is based on process-based morphodynamic modelling of an idealized tidal basin. Compared to a similar previous study (Dissanayake et al., 2012) the model

is extended by including multi-fraction sediment transport, viz. three fractions (sand, silt & clay) instead of a single sand fraction.

1. However, instead of focusing on a single real-world tidal inlet system (Ameland Inlet in the Dutch Wadden Sea) by the former study the present manuscript claims that the findings are relevant for mixed-energy tidal inlet systems that occur throughout the world. I think that this is a too far-reaching claim, as the model simulations concern only one idealized tidal inlet.

Response: While the system we study is a single inlet system, the system geometry, and physical characteristics represent mixed energy systems, and as outlined in Table 1, there are many similarities of the system studied to many other systems worldwide. As shown in Table 1, this type of barrier island coast occurs throughout the world. Moreover, this type of backbarrier morphology and geometry tends to produce ebb-dominated inlet channels. Exceptions to this characterization occur where there have been significant human alterations (e.g., dredging at Saint Mary's Inlet, GA; tidal basin shortening by dam constructions along The Netherlands coast).

2. Thus, no variation in basin geometry, initial bathymetry, initial bottom composition, and forcing conditions is considered. It is well known that the response of a tidal basin to (changing) sea level rise, e.g., whether (accelerating) sea level rise causes extra import or extra export, very much depends on its morphology (Friedrichs et al 1990). The modelling study as presented in the manuscript can at most be relevant for a certain group/category of tidal inlets in the world. However, it is not clear which tidal inlets the modelled case would represent, as some essential characteristics of the idealized basin are not presented. It would help if the hypsometry of the basin was presented.

Response: We have included the basin hypsometry in the SI. However, it is beyond the goals of this paper to investigate all the possible backbarrier variability as listed by Reviewer 2. These are admirable goals and hopefully will be pursued by us and other investigators in future studies. We chose to limit our study to standing wave systems that produce ebb-dominant inlet channels so we could investigate how different grain sizes will respond to backbarrier evolution forced by SLR. We added a statement in the paper acknowledging that variability in backbarrier geometry and hypsometry could and likely would vary results from this conceptualized model. (see: Figure S2)

3. Moreover, the modeled case does not represent mixed-energy tidal inlets as waves are ignored in the model.

Response: We are investigating the evolution of the backbarrier systems as a consequence of SLR. Because these basins are characterized by marsh and tidal creeks with limited open-water area, we do not consider waves an important forcing agent. Wave energy is important in affecting the morphology of the ebb-tidal delta and the inlet shoreline but plays a very minor role in hydrodynamics and sediment transport in the backbarrier. Also, the sand volume of the ebb-tidal delta is primarily a function of the tidal prism as noted by Walton and Adams, 1976.

4. Another concern of mine is that no validation of the used model has been presented. Not any comparison of the model results after the spin-up simulation, which are used as initial conditions (bathymetry, and bottom composition?) for the simulations with sea level rise, with any real-world case is presented. Even though the idealized case cannot be compared with any specific case, at least some essential morphological and sedimentological characteristics should be qualitatively compared with real-world cases.

Response: We have added comparisons of model results to the Plum Island system and added this to the SI. Basically, we show that Plum Island Sound has a geometry consistent with that of modeled basin and experiences standing tidal wave signature. We also

demonstrate that like the modeled basin sand is imported through the inlet and deposited on a large backbarrier sand shoal known as Middle Ground Shoals (flood-tidal delta).

5. One of the findings that the fine sediment fractions, especially the clay fraction sediment, are exported due to sea level rise is against the finding from a recent study for the Wadden Sea based on field observations (Colina Alonso et al., 2021). I am afraid that the finding in this modelling study is due to some specific setting concerning e.g., the bottom composition.

Response: We must take exception to this comment. We had considered the Alonso et al (2021) study, but it was noted in the paper that the import of clay was not attributed to SLR, but rather to closure of a major part of the basin. The following quote is taken directly from the Alonso et al 2021 paper: “Elias et al. (2012) show that the sedimentation of the basins has been primarily a response to the closure of the South Sea and not an adaptation to SLR.”

6. The conclusion that “Sediment sinks on the ebb-delta and within the backbarrier system will be at the expense of proximal sediment reservoirs” is not based on the presented modeling results. As waves are not included in the model, the model results do not show sediment transport from the adjacent coasts to the ebb-tidal delta.

Response: The modeling results show that ebb-tidal delta grows in volume. Movement of sand into tidal inlets from adjacent barrier shore is a very well know concept dating to the 1970s along barrier coasts throughout the world (Dean and Walton, 1977; Oertel 1977; Findlay; 1978; Hayes, 1979; Nummedal and Penland, 1981; FitzGerald 1984, Smith and FitzGerald, 1994; FitzGerald et all 2000; Elias van der Spek, 2017; Herrling and Winter 2018; Elias et al 2022). These publications show diagrams with arrows indicating longshore sediment transport into the inlet. Thus, we did not believe it was necessary to impose waves into the model to affirm what is established in the literature.

7. The manuscript does not include all essential information concerning the model set up.
 - a. Which (transport /erosion) formulations are used for the three sediment fractions?

Response: This is now stipulated.

- b. How do you distinguish intertidal flat and marsh platform (or are they the same)?

Response: We added elevation range (MSL to MHW) where vegetation was imposed.

- c. Unless the model files are made available the presented information does not allow someone else to reproduce the presented modelling results.

Response: The simulation and set-up files have been uploaded.

Details

Line 71-73. Also, for fine sediment maximum current asymmetry has influence on the net residual flux in addition to the asymmetry in slack water duration. Which of the asymmetries is more important differs from case to case.

Response: Agreed, this has been corrected in the text

Line 79-82. The statement on the relation between slack water duration asymmetry and direction of net transport is wrong. The correct relation is: longer pre-ebb / HW slack promotes import of fine sediment, and vice versa.

Response: We apologize for this error and have changed the text. Thanks for catching this!

Line 92-93: what is meant by “the flood tidal wave”? Highwater? Or rising tide?

Response: Changed to “flooding tide”

Line 108-109. The predicted decreases of the tidal flat volume and ebb-tidal delta volume are not due to erosion but due to the rising sea level. The volume of intertidal flat in the basin is defined as the sediment volume between MOVING LW and HW. More recent results on impact of sea level rise on the Dutch Wadden Sea are reported by Lodder et al. (2022) and Huismans et al. (2022).

Response: In addition to the Van Goor, M. A., Zitman, T. J., Wang, Z. B. & Stive, M. J. F. Impact of sea-level rise on the morphological equilibrium state of tidal inlets. *Mar Geol* **202**, 211–227 (2003) reference has been added: Wang, Z.B., Elias, E.P.L., Van der Spek, A.J.F., Lodder, Q.J., 2018. Sediment budget and morphological development of the Dutch Wadden Sea - impact of accelerated sea-level rise and subsidence until 2100. *Neth. J. Geoscience*. 97–3, 183–214, which discusses import of sand to tidal basins through contribution from ebb-tidal deltas. In addition, we will cite Lodder et al 2022 and Huismans et al 2022 as studies that show that demand of sediment by tidal basins is a response to SLR.

Line 210. What is “cumulative transport rate”? Cumulative transport often (e.g., in the output of Delft3D) refers to the time integral of sediment transport rate, but in Fig.4c it has the unit of sediment transport rate.

Response: The plotted quantities represent the time derivative (slope) of the cumulative transport (Delft3D output that is the time integral of sediment transport rate, units of volume, after it has been filtered to remove tidal fluctuations) so properly have the units of volume/time. A better description is “Residual Transport Rate”, and figure labels and captions are updated accordingly.

Line 292-293. The tidal flat above MSL vegetated? To my knowledge, the transition between bare mud flat and vegetated marsh is around MHW. I do understand that the used model only includes a single tidal constituent, so MHW cannot be used.

Response: We have added the McKee and Patrick, 1988 citation (see Figure 2) that shows that most marshes begin colonizing tidal elevation at mean sea level and above.

Line 327. 911?

Response: Should be 9 and 11

Line 396. The open boundary? There are three boundaries in the sea.

Response: This has been updated to read “offshore open boundaries”

Line 397. The tidal wave also deforms as it propagates in the shallow sea outside the inlet.

Response: We added the clause “distortion of the tidal wave over the nearshore zone between the offshore boundaries and tidal inlet is negligible”

Line 456-458. Waves also influence residual sediment transport as they can cause sediment concentration difference between inside and outside the inlet.

Response: We acknowledged the importance of waves in modifying the ebb-tidal delta and resuspending fine-grained sediment in some nearshore settings (e.g., Castagno et al 2018), which might affect basinal sediment exchange with the coastal ocean, but this paper is focused primarily on the backbarrier basin.

References added.

Castagno, K. A., Jiménez-Robles, A. M., Donnelly, J. P., Wiberg, P. L., Fenster, M. S., & Fagherazzi, S. (2018). Intense storms increase the stability of tidal bays. *Geophysical Research Letters*, *45*. <https://doi.org/10.1029/2018GL078208>

Elias, E.P.L., Van der Spek, A.J.F., Wang, Z.B. & De Ronde, J.G., 2012. Morpho- dynamic development and sediment budget of the Dutch Wadden Sea over the last century. *Netherlands Journal of Geosciences / Geologie en Mijnbouw* 91: 293–310.

Wang, Zheng Bing, Elias, E.P.L., van der Spek, A.J.F., Lodder, Q.J., 2018. Sediment budget and morphological development of the Dutch Wadden Sea: impact of accelerated sea-level rise and subsidence until 2100. *Netherlands Journal of Geosciences* 97 (3), 183–214. <https://doi.org/10.1017/njg.2018.8>. Cambridge Core.

Friedrichs, C.T., Aubrey, D.G., Speer, P.E., 1990. Impact of relative sea level rise on evolution of shallow estuaries. In Cheng R.T. (ed.), *Residual current and long-term transport*, Springer-Verlag, New York.

Huismans, Y., van der Spek, A., Lodder, Q., Zijlstra, R., Elias, E., Wang, Z.B., 2022. Development of intertidal flats in the Dutch Wadden Sea in response to a rising sea level: Spatial differentiation and sensitivity to the rate of sea level rise. *Ocean and Coastal Management* 216, 105969. <https://doi.org/10.1016/j.ocecoaman.2021.105969>.

Lodder, Q., Huismans, Y., Elias, E., de Looft, H., Wang, Z.B., 2022. Future sediment exchange between the Dutch Wadden Sea and North Sea Coast - Insights based on ASMITA modelling. *Ocean and Coastal Management* 219, 106067. <https://doi.org/10.1016/j.ocecoaman.2022.106067>.

Pearson, S. G., van Prooijen, B. C., Elias, E. P. L., Vitousek, S., & Wang, Z. B. (2020). Sediment connectivity: A framework for analyzing coastal sediment transport pathways. *Journal of Geophysical Research: Earth Surface*, 125, e2020JF005595. <https://doi.org/10.1029/2020JF005595>

REVIEWER #3 (Remarks to the Author):

General:

The manuscript is generally well written, and of high quality.
I really like the detail and thought put into the beginning of the discussion.

1. Unfortunately, the chosen main message is not always clear, which is reflected in inconsistencies in the manuscripts structure.

Response: We have made substantive changes to the Abstract, Introduction, and Conclusions to ensure that our main message is focused and clear.

More specifically:

2. It is unclear whether the focus lies in the morphological backbarrier basin response to SLR or the interaction between vegetated and unvegetated platform in response to SLR.

Response: The focus of the paper is the morphological evolution of the backbarrier as forced by SLR, which causes an overall deepening of the basin and increase in tidal prism. This evolutionary path led to the ensuing goal of determining how the changing hydrodynamics of the basin affected the import versus export of different grain sizes. Sediment being transported in the model was derived from sediment being eroded in the backbarrier as well as sediment outside of the basin. Tracking was accomplished by stipulating a bed composed of clay, silt, and sand.

3. It is unclear whether the main question focuses on the impact of multiple grainsizes on the basin response and how applicable this would be to other systems?

Response: Yes, a major goal is to show how net sediment transport trends of individual grain sizes (clay, silt, sand) will evolve as SLR changes the hydrodynamics of the backbarrier. A unifying characteristic of Mixed-energy barrier coasts throughout the world is that backbarriers are dominated by sandy substrates near the inlet mouth and grain size tends to fine up tidal creeks and up basin. Thus, the trends illustrated here are transferable and relevant to other Mixed-energy tidal

inlet systems. For example, the sand moving into the basin is similar to what is documented in field studies by Wang et al 2018: “A characteristic feature of the Wadden Sea region is its continuous sedimentation of the tidal flats in order to keep pace with relative SLR, and its siltation along the Wadden shores. These processes are responsible for an important influx of sand, which is basically delivered by the adjacent coastal system. This is the cause of a structural retreat of the Wadden island shores.”

4. How important is the expansion of the ebb-tidal data, how is this incorporated in the model?

Response: One of the important responses to SLR is a deepening of the basin and an attendant increase in tidal prism. This explains why the volume of the ebb-tidal delta increases (Walton and Adams, 1976). Our modeling captures this dynamic process very well effectively differentiating between the control case and the remaining scenarios with sea level rise.

5. I am not saying these sub-questions could not be linked, however in the current form this link remains unclear to the reader.

Response: Understood and thus the major question and sub-questions are more intimately linked in Abstract, Introduction and the cascading and connected responses are better defined in the Discussion, & Conclusions.

6. Additionally, what is the main message the reader can take home from this paper, the current manuscript structure more closely reflects a submission to a specialized journal like JGR than nature communications?

Response: This study provides a major advancement in our understanding of how Mixed Energy backbarriers will respond to SLR. Our findings show that basins will import sand and increasing tidal prism enlarges the ebb-tidal deltas. Also, fine-grained sediment is exported which diminishes inorganic sediment that can help maintain the marsh platform. These are very important findings because they address backbarrier evolution and the future resiliency of the marsh .

7. **Broader Impact:** the manuscript portrays the Results to be generally applicable to backbarrier salt marshes, however it remains uncertain how representative the physical- and boundary conditions of the chosen test case (i.e. Plum Island Sound) is for barriers worldwide. For instance, even along the US East Coast we encounter a wide variety of backbarrier basin sizes and forcing conditions (mixed, mixed-tide, wave-dominated) at the seaward side. For instance, mixed energy systems as indicated in Table.1 have vastly different dynamics in barrier inlet and thus sediment fluxes, for instance Wadden Sea Inlets act as sinks for littoral sediment transport except sediment transported by sediment bypassing around the ebb-tidal delta.

Response: We acknowledge that the coasts listed in Table 1 have different hydraulic regimes and that is now better stated in the paper. However, we are modeling a conceptual basin that is fashioned after Mixed-Energy barrier island coasts (sensu: Hayes 1979) in which the backbarrier contains a relatively short basin length that produces a standing tidal wave. All the coasts listed in Table 1 have a tidal basin of similar length to that of the model and support standing tidal waves. Likewise, all these coasts (Table 1) have semidiurnal tides and their inlets experience inlet sediment bypassing and variable sediment fluxes through the inlet mouth. Present sediment dynamics of the Wadden Sea Inlets are largely a response to basin infilling and artificial closures (Elias et al., 2021) so are less comparable.

8. Focusing on the specific model case study, how well is the model capable reproducing the hydrodynamic characteristics of the model system, e.g., tidal asymmetry, slack water duration. How well is the model, on average able to reproduce the distribution of different sediment types as shown in Fig.1, finer (clay silt) and coarse (sand) sediment types?

Response: We have chosen to add Plum Island Sound in the **SI** as a comparison to the conceptual model in terms of basal geometry, standing wave character (as demonstrated with velocity times series) and importation of sand.

9. Vegetation: I appreciate the sensitivity study carried out by the authors, testing the impact of vegetation presence on the platform. It is not fully clear which vegetation parameters have been tested and where on the platform they were implemented? section updated to include descriptions of vegetation model, parameters, and marsh platform definition: “The impacts of vegetation on the flow across the marsh surface and resulting influence on sediment transport patterns are neglected in the model experiments presented herein. Additional sensitivity analyses were performed that utilize the Delft3D trachytopo roughness schematization (as a function of vegetation stem height and density according to the *Baptist 2* model ⁷⁸) that increases bed roughness over the marsh platform ⁷⁸, defined as the basin areas with elevations between the time-varying MSL and MHW elevations.”

10. **Abstract:** is not easy to understand and fails to summarize the main points.

Response: The Abstract has been completely rewritten to simplify and emphasize the major points and findings of this study.

More Specifically:

Line 21: from the abstract the effect of imposed salt marsh accretion is unclear, which trend gets diminished, the shift from export to import?

Response: We have made this clear in the Abstract that when we impose marsh accretion, the trend of coarse silt and sand influx decreases to a single 40-year period. After this period, coarse silt and sand are exported.

Line 22: What do the authors mean by the trend for sand is more subdued? How is does clay import or export change with or without saltmarshes?

Response: We meant that the curve for sand was flatter and the trend less dramatic than that for coarse silt. We have changed the wording to make this point clearer.

Line 23: "This loss of sand will threaten barrier resources and infrastructure" remains unclear

Response: The modeling Results show that marsh will be converted to open water deepening the backbarrier and increasing tidal prism and corresponding increase in volume of the ebb-tidal delta. It has been established that change in volume of the ebb-tidal delta will influence inlet sediment bypassing and thus sand reservoirs of adjacent barriers (Pearson et al 2020; added to references). We revised the paper for clarification.

Main text: generally difficult to follow

Response: The Introduction has been rewritten to correct these issues.

line 33: rephrase "the ~37% of coastal population that live there"

Response: We have changed this to read: ... particularly for the ~10% of the global population who in coastal areas less than 10 m above sea level²

line 37: what are the authors referring to, what type of coastal systems do they focus on?

is this true for all systems,.. rocky and sedimentary coasts? cliff erosion and dune overwash might appear at a specific type of coastal systems, however are they the main driver of coastal hazards?

Response: This text has been rewritten.

Line 40: are the barriers themselves heavily developed, which systems do the authors have in mind?

or are the authors referring to the back-barrier basin? Before this line, for the reader to appreciate the subject of this study a short introduction in barrier-systems needs to be given, what kind of morphological units can we distinguish, how can they structure barrier dynamics?

Response: This section has been completely rewritten, including the addition of a primer for what a “Mixed Energy” Barrier system is.

Line 45: What do field observations reveal? Is this proposed gap a problem with our ability to model barrier systems or a trend also observed across real systems?

The proposed summary is very difficult to follow:

"Some studies predict that tidal inlets will import sediment 8–10 or that tidal range will control net inlet transport patterns 11, and more recent studies suggest that sediment will be exported as sea level rises 12,13, retarding the ability of marshes to build vertically".

Response: This section has been completely rewritten to sharpen the difference of opinions in predicted SLR forcings of future sediment transport trends and the focus of our paper. Additional citations have been added, including field examples.

 how is this sentence linked to the underlying processing increasing vs. decreasing tidal prism, increasing vs. decreasing load of suspended or bed load sediment,.. are there also rivers discharging sediment into the backbarrier lagoon? what is the impact of waves, vs. tides, which process is dominant, when and where?

Response: We have removed this from the Introduction and put a fuller treatment of future increases in tidal prism and its effects in the Discussion. We did not include backbarrier systems that contain important sediment contribution.

Line 49: the research question is unclear ?? in general backbarrier lagoons are characterized by finer sediment types, such as fine sand, silt, and clay. Is fine sediment considered? What do the authors mean by sequestration of sand?

Response: Again, this section has been rewritten and effects of tidal prism increases and sequestration of sand on ebb tidal deltas have been removed and in put in the Discussion.

Line 52: Especially flood tidal delta evolution is governed by waves by littoral drift (longshore transport) and sediment bypassing caused by oblique coastal waves!  please revise, .e.g. De Swart, H. E., and J. T. F. Zimmerman. "Morphodynamics of tidal inlet systems." Annual review of fluid mechanics 41.1 (2009): 203-229. what kind of barriers are the authors referring to, tide-, mixed- or wave-dominated barriers since the relative contribution of processes will vary as will ebb and flood-tidal delta volumes, shapes and dynamics? e.g. using the classification of Hayes 1979

Response: We are intimately familiar with Hayes’ classification (one of us was his PhD student and helped with his 1979 paper, particularly Figure 15 of that paper). Barriers only exist in Wave-dominated and Mixed Energy settings. Ebb-tidal delta morphology can be related to waves and tides and Hubbard et al 1979 came up with a classification: Tide- Dominated, Wave-Dominated and Transitional to describe ebb-tidal delta shape. One of us has also published numerous papers concerning sediment transport in the vicinity of tidal inlets. Most importantly, this section of the introduction has been removed and the important points have been added to the Discussion.

Line 58/supplement: I recommend the authors go over the supplementary text and remove present typos: Moreover, the discussion on the marsh platform keeping pace with sea level rise, seems to jump between an oversimplified 0d approach(1st paragraph) and a more detailed/realistic marsh platform, levees and channel approach (2nd paragraph).

Response: We apologize for the typos, which have been corrected. The structure of this section was intentional, as marsh platform elevation dynamics are a “0d” process governed by factors such as sediment supply, inundation duration, and organic production that vary spatially.

Suggest for the authors to:

1st merge this to points-of-view in a cohesive text

2nd since salt marshes seem to exert important control on the backbarrier system in respect to the posed research questions I propose to move at least part of it to the main text!

Response: We agree and have made this an important point (see below).

Line 62-85: is an excellent paragraph and in my opinion should be moved into the main focus of the current study

Response: We agree and have moved this paragraph up into the Main focus of goals.

Line 98: Will the net import of sand into the inlet lead to different cross-sectional areas in tidal channels and shallower backbarrier basin depths? Will sand be also transported onto the marsh platform? Is an increase in transport capacity to be expected leading to coarser particles being deposited on intertidal platforms. If so, has this been observed in previous field studies?

The provided references mainly point to a model study (Friedrichs and Aubrey 1988), and a review from FitzGerald 2008)?

Response: We have removed the sentence discussing sand imported into the basin from the barrier shoreline but have added references to demonstrate field evidence of sand import to the backbarrier by flood- dominated channels (Nowacki and Ogston 2013; FitzGerald et al 2022).

Line 122- 138: different impressions on marsh survival and ebb-flood dominance originate from field and model exercises with varying complexity? I think here the authors miss a chance to justify the current approach, i.e., what do we know from field studies what are their boundaries, what do we need models for (and what are their boundaries):

Response: We have added the clause “and investigate longer timescales and more alternative SLR conditions than possible with field studies”

Line 138: current research question

Line 139: I think using a numerical model does not represent a theoretical investigation, or are the authors referring to something else?

Response: We have replaced “theoretical” with “numerical”

Line 140-143: here the research gap and question remain unclear and the reader is provided with details, which are appreciated but unfortunately without sufficient context.

Line 143: "Thus we allow for full.." unclear compared to what kind of model?

Response: Ioannis will rewrite these sentences

Line 147: Table.1 incomplete how does sediment fluxes and tidal ranges compare?

Response: Information about sediment fluxes for the different sites listed in Table 1 largely does not exist. Also, this study is not concerned with how sediment fluxes are influenced by tidal range. That is beyond the goals of the paper.

Line 178: reference missing

Response: Citations have been added

Line 223-224: which figure shows slack-tide asymmetry?

Response: This has been changed Please see Figure 6C.

Line 229: rephrase

Response: I am unsure what sentence he wants us to change? Ioannis and Kevin help!

KH: maybe the “after 200 years and approximately 1 m of SLR”? Could rephrase to “but at the end of the simulation with 1 m of cumulative SLR, residual transport ...)”

Line 239-288: is an excellent discussion which should be put more in the focus of this study

Response: This section is actually well-positioned in the paper because it lays the foundation (see second sentence) for the immediate next section of the paper: “Changes in flood/ebb asymmetry and resulting net sediment transport”

Line 253: I think the authors mean infilling of the channels which could be stated more explicitly

Response: mean depth at low tide can decrease due to both channel infilling and the increased inundation extents at low tide (even with a static channel bed). If the elevation of low tide continues to increase, channel banks can begin to be inundated even during low tide, introducing a larger area low-depth values that decrease the mean depth.

Line 298: I am not sure the reference to Aubrey Friedrichs is applicable here

Response: We did not cite an Aubrey reference in this section of the paper. The Friedrichs & Aubrey reference is cited on Line 248, as this paper discusses ebb dominance as a function of tidal amplitude and channel depth.

Discussion:

Line 313 if the Main aim was to disproving results by Donatelli 2020, and thereby indicating a positive feedback between marsh loss and sediment trapping efficiency this should have been indicated earlier?

Response: The Introduction has been rewritten and the goals of the paper have been narrowed and more clearly stated. The Donatelli et al 2020 paper and Zhang et al 2020 suggest that SLR will result in the export of sediment which is contrary to other studies, and thus one aspect of our paper serves to investigate this divergence of ideas. This is now better defined in the Introduction.

Line 398: was the bathymetry in some dynamic equilibrium? unclear using Fig.3 and Fig.4c?

Response: Additional information has been provided in this sentence for clarity: “While a 200-yr morphologic timescale simulation is inadequate to achieve full dynamic equilibrium with no further bathymetry changes⁴¹ (see Figure 2a), both the volume of backbarrier morphologic features (see Figure 3) and net sediment fluxes (see Figure 4c) have reached steady-state conditions (indicating dynamic equilibrium of aggregate morphology) onto which perturbations could be imposed.”

Line 410: how comparable are the forcing compared to your system as in sediment supply and tidal range,... which might alter their dynamics as shown in line 244ff.

Response: They are comparable to Mixed-Energy systems listed in Table 1. The lack of significant sediment input from rivers or other upland sources is similarly consistent with many (though not all) of the prototype basins.

Ebb-tidal delta volume is dependent on wave and tide dominance, which raises the question to how many systems Plum Island is comparable to?

Response: Ebb-tidal delta volume is primarily dependent on tidal prism and as Walton and Adams (1976) showed, wave energy has a secondary influence on volume. The morphology of the ebb delta indeed is a product of wave versus tidal dominance, but not the volume. The Plum Island Sound ebb delta exhibits a moderately ebb-dominant morphology.

Line 419: see point Line 410

Response: see response to Line 410 comment above.

Line 446: specifics of the model setup are missing, sediment transport equations, treatment of mixed beds, if not in the main article this should be present in the Appendix so the study stays reproducible.

Response: This section is updated to note sediment transport equations and more detail on initial bed composition

Line 463: It is unclear how the MEM implemented, I understand that not the whole model can be described, however sufficient information should be present to interpret the results, how does accretion rate change over space?

Response: More information is given on the mechanisms of the MEM: "...such as the Marsh Elevation Model (MEM) that computes accretion rates based on biomass production and inorganic sedimentation". For more information on the specific model formulations and settings in the referenced study, see Farron et al., 2020.

Line 471: insufficient details describing vegetation roughness tests.

Response: This section is updated to include descriptions of vegetation model, parameters, and marsh platform definition: "The impacts of vegetation on the flow across the marsh surface and resulting influence on sediment transport patterns are neglected in the model experiments presented herein. Additional sensitivity analyses were performed that utilize the Delft3D trachytopo roughness schematization (as a function of vegetation stem height and density according to the *Baptist 2* model⁷⁸) that increases bed roughness over the marsh platform⁷⁸, defined as the basin areas with elevations between the time-varying MSL and MHW elevations."

REVIEWER COMMENTS

Reviewer #2 (Remarks to the Author):

I believe that the authors did a good job in processing the comments from the reviewers and improved the manuscript. However, there are still a couple of issues remaining, which related to the rebuttal of the authors to my comments (Reviewer #2):

Rebuttal #5: It is true that the effect of the closure has been more important than that of SLR in the western Dutch Wadden Sea. However, both the closure and SLR cause sediment demand in the basins. The data analysis shows that the contribution of mud to sediment import became more important in time while the relative importance of the closure to SLR decreased. Moreover, tidal basins in the Wadden Sea which are not affected by closures, e.g. the basin of Ameland Inlet, have also been importing mud. Are there observations from any system that support the finding that SLR causes fine sediment export and coarse sediment import? Can the authors elaborate about the possible effects of the initial sediment composition setting in the model on this?

Rebuttal #6: My comment was apparently not clearly formulated. I was not saying that I disagree with the statement. I was saying that the statement was not (directly) based on the model results. I think it is appropriate to make clear if the conclusion is based on model results or based on further reasoning using literature. The formulation in the manuscript suggests to me that the conclusion is a direct outcome of the modelling work.

Rebuttal to my comment on Line 71-73: By comparing two versions of the manuscript it seems to me that the formulation has not been changed, in contradiction to what the rebuttal says.

Reviewer #3 (Remarks to the Author):

I appreciate the work to authors have put in compiling the revised version.
The main message of the paper now is much clearer.

Unfortunately there are some concerns which prevent me from fully appreciating the provided results.
Remaining concerns:

- applicability of the provided results to other mixed energy systems
- impact of initial condition and added sediment on the marsh platform on predicted results
- missing information regarding the model setup

General comments:

1. I appreciate the change that has been made to the Abstract, Introduction and Conclusions of the manuscript.

2. If the main focus lies on how the morphodynamics of the backbarrier lagoon is affecting the net sediment transport of different sediment classes through the inlet, I would assume the change in tidal inlet dimensions, and sediment supply would be of high importance. For that reason I was wondering why waves acting offshore and at the inlet, shaping the ebb-tidal delta have been omitted ?? If the focus would be mainly on redistribution of sediments within the backbarrier lagoon, I agree that modelling waves can be omitted. How does the ebb-tidal delta volume compare with other mixed systems??

3.

a.. Since the focus lies on multiple-grain-sizes and their distribution over SLR.

How do the authors motivate their choice of the initial stratigraphy, 43% fine sand, 43% coarse silt and 14% clay? Is this specific for the current system?

How important are these initial conditions for the predicted sediment transport patterns over SLR??

How was the interaction between different sediment fractions dealt with in the model?

When are sediments within a mixed active layer top cell of the bed transported as cohesive or non-cohesive material ?

Which transitional coefficient did the authors use to delineate cohesive-, from non-cohesive beds, e.g. the default, why ?

This needs to be added and motivated in the method section.

Would the authors expect that this coefficient influences the robustness of their model predictions?

b.. This trend does not illustrate that the results derived (i.e. for the impact of SLR) from the model will be applicable to other Mixed-energy systems, see comment below: see point.10

4. See comment above, and the impact of the ebb-tidal delta volume to wave forcing.

Previous studies have shown that ebb-tidal delta volume in mixed energy systems can be influenced by waves,

e.g. Lenstra, Klaas JH, et al. "Cyclic channel-shoal dynamics at the Ameland inlet: the impact on waves, tides, and sediment transport." *Ocean Dynamics* 69 (2019): 409-425.

How would waves influence the predicted morphological development over SLR ?

5. I appreciate the revised version, leading to a much more streamlined link between the sub-questions

6. Although the system is a mixed-energy system, that does not necessarily mean results derived from this geometric model setup (presented in table.1) are representative of other mixed energy systems,..

Can derived results be applied to Frisian island systems or other mixed energy systems along the US east coasts ? I have major doubts about that. To make such a claim the robustness of the results we need to be evaluated varying geometry, bathymetry, initial sediment composition and

wave/tide forcing ratios(within the mixed energy range),... which is beyond the scope of the current study.

I agree the loss of fine sediments would potentially counteract salt marsh survival over sea level rise, this is in itself would be a relevant finding for the plum island lagoon. I suggest that the authors explain why wave dynamics are less important at the current system, and then focus on the wetland fine sediment loss message.

However, extrapolating results to other barriers, is in my opinion not possible, without a significant investment in add. model scenarios.

7. As indicated by the authors, all the inlet's listed in Table.1 experience inlet sediment bypassing, pointing towards the importance of wave-induced long-shore transport around the ebb-tidal delta and thus wave processes. I understand that the author's are modelling a conceptual basin, which I think is a valid approach, however the question remains whether all essential processes(specifically waves) have also been accounted for?

8. I want to thank the authors for pointing my to the SI, showing the standing wave character and import of sand. However SI.2 and 3 are model results, are there any indications e.g. field observations and statistical relationships by managing authorities that the predicted rates are in the right order of magnitude? Was their any validation carried out? for this or other comparable systems, .e.g scaling-relationships?

9. Which parameters have been used for the Baptist equation, more specifically, stem diameter, stem height, stem density and drag coefficient.

This information is crucial for the reproducibility of the study.

10. I appreciate the updated abstract.

Unfortunately it remains unclear how sediment accretion was imposed, i.e. what sediment classes where added.

Line 413-430 and Line 450-460?

Did the authors simply increase the elevation o the most top-layers, including the active layer and all mixed sediment classes or did they add a specific size-fraction?

Is any portion of the accreted sediment part of sediment transport? Will this affect the resulting sediment balance, e.g. with marsh is converted to open-water?

Detailed suggestions:

I appreciate the work the authors have put into the revised version and the letter to the reviewer, and responding to my suggestions.

However at sever points it seems that resubmission might have been premature, still containing comments intended for within the author-group?

This could have been avoided.

e.g. Line 229: rephrase

Response: I am unsure what sentence he wants us to change? Ioannis and Kevin help!

Moreover, whereas most of the comments have been answered satisfactory.

Specific responses still did not provide sufficient information, which threatens the reproducibility:

e.g. Line 446: New Line 446, e.g. the Parteniades-Krone equation is used to fine sediment transport, which crit. shear stress for erosion and erosion rate parameter has been used.

There should be a table in the appendix, stating the chosen model setup?

e.g. Line 463: how was the MEM model implement, see comment above what sediment classes have been added ?

e.g. Line 471: again the specific parameter settings are missing which should be provided in some table in the appendix

REVIEWER COMMENTS

Reviewer #2 (Remarks to the Author):

I believe that the authors did a good job in processing the comments from the reviewers and improved the manuscript. However, there are still a couple of issues remaining, which related to the rebuttal of the authors to my comments (Reviewer #2):

Rebuttal #5: It is true that the effect of the closure has been more important than that of SLR in the western Dutch Wadden Sea. However, both the closure and SLR cause sediment demand in the basins. The data analysis shows that the contribution of mud to sediment import became more important in time while the relative importance of the closure to SLR decreased. Moreover, tidal basins in the Wadden Sea which are not affected by closures, e.g. the basin of Ameland Inlet, have also been importing mud. Are there observations from any system that support the finding that SLR causes fine sediment export and coarse sediment import? Can the authors elaborate about the possible effects of the initial sediment composition setting in the model on this?

Response: There have been several recent studies reporting sediment import to backbarrier basins including inlets along the Virginia coast (Castagno et al 2018) and at New Inlet in Central Massachusetts (Baranes et al 2022). The import of sediment through Rockaway Inlet into Jamaica Bay along the Long Island coast in New York was published by Renfro et al (2010) and later confirmed by Donatelli et al (2020). The response of the model to the initial bed sediment composition is also referenced in another comment by the reviewer. To address both comments/questions, we have performed additional experiments where we varied the initial bed composition by increasing and decreasing the mud content and repeated the control simulations as a sensitivity analysis. We found that varying the clay content (4%, 14, 24%) did not change in sediment transport trends (export and import) or magnitude of the sand and silt fractions. Likewise, clay is continuously exported and as expected the larger the percentage of clay sized sediment, the greater amount clay is cumulatively transported (exported).

Rebuttal #6: My comment was apparently not clearly formulated. I was not saying that I disagree with the statement. I was saying that the statement was not (directly) based on the model results. I think it is appropriate to make clear if the conclusion is based on model results or based on further reasoning using literature. The formulation in the manuscript suggests to me that the conclusion is a direct outcome of the modelling work.

Response: Thank you for pointing this out, indeed our conclusion is an outgrowth of the modeling results, but also based on inlet processes published in the literature. We have made this clear in the text:

Changed to:

“Contrary to recent studies of mixed-energy tidal systems, particularly in the Netherlands⁹, results from this study infer that the ebb-tidal delta will expand with SLR due to increasing tidal prism¹⁷, which may reflect the use of a moderate-sized tidal inlet-basin system for the modeling. Sediment losses to the backbarrier system and sand trapped on the ebb-delta and will lessen inlet sediment bypassing¹⁸ and reduce the proximal sediment reservoirs, most likely from the adjacent barriers. This will make some barriers more susceptible to breaching and transgression.”

Rebuttal to my comment on Line 71-73: By comparing two versions of the manuscript it seems to me that the formulation has not been changed, in contradiction to what the rebuttal says.

“Also, for fine sediment maximum current asymmetry has influence on the net residual flux in addition to the asymmetry in slack water duration. Which of the asymmetries is more important differs from case to case.”

Response: We appreciate the comment; it seems we omitted revising the text. We edited the manuscript for clarity and to reflect the following message: current asymmetry dominates bedload transport, and duration asymmetry for slack water dominates suspended load transport.

Reviewer #3 (Remarks to the Author):

I appreciate the work the authors have put in compiling the revised version. The main message of the paper now is much clearer.

Unfortunately, there are some concerns which prevent me from fully appreciating the provided results.

Remaining concerns:

- applicability of the provided results to other mixed energy systems
- impact of initial condition and added sediment on the marsh platform on predicted results
- missing information regarding the model setup

General comments:

1. I appreciate the change that has been made to the Abstract, Introduction and Conclusions of the manuscript.

Response: Thank you

2. If the main focus lies on how the morphodynamics of the backbarrier lagoon is affecting the net sediment transport of different sediment classes through the inlet, I would assume the change in tidal inlet dimensions, and sediment supply would be of high importance.

For that reason I was wondering why waves acting offshore and at the inlet, shaping the ebb-tidal delta have been omitted? If the focus would be mainly on redistribution of sediments within the backbarrier lagoon, I agree that modelling waves can be omitted.

How does the ebb-tidal delta volume compare with other mixed systems?

Response: The focus of the paper is indeed on the morphodynamics of the backbarrier as it changes with rising sea level. One of the important outputs of the modeling is that the tidal prism increases, which, as the published literature would predict, results in an increase in dimensions of the inlet and enlargement of the ebb-tidal delta. These relationships are shown in our Figure 2. Both inlet size and volume of the ebb-tidal delta are related primarily to tidal prism. Waves impart a much lesser influence as many studies have demonstrated, although the reviewer is correct in saying that wave versus tidal energy will affect the shape of the ebb delta. Moreover, most of the sediment exchanged between the backbarrier basin and the coastal ocean occurs through the main channel, which due to its depth, is primarily affected by tidal currents and not waves. The computed volume of the ebb-tidal delta adheres well with the Walton and Adams (1976) relationship and is similar in planform to many mixed-energy ebb-tidal deltas.

3.a. Since the focus lies on multiple-grain-sizes and their distribution during SLR.

How do the authors motivate their choice of the initial stratigraphy, 43% fine sand, 43% coarse silt and 14% clay? Is this specific for the current system?

How important are these initial conditions for the predicted sediment transport patterns over SLR??

Response: We chose to use a relatively low clay content for our sediment regime because most tidal inlets tend to be coarser-grained systems (sand and coarse silt) but may contain clay at the heads of smaller tidal creeks and up estuary (at the drainage divides in East Frisians). However, close to the inlet and main tidal channels, where most of the sediment is exchanged, are floored by sand, as are the flood

deltas and point bars. As channels deepen in the backbarrier, they may erode into sedimentary layers containing more clay, but usually in mixed energy systems that contain backbarrier marshes and tidal creeks or tidal flats (East Frisian Islands, North Frisian Islands, Copper River Delta barrier, Alaska), the channels erode into deposits that consist of sandy sediment that was transported into the backbarrier as the barriers and inlets were stabilizing and the basin was filling. Again, in these cases, clay will be minimal. However, to look at the importance of clay, we have run the model using a content of 10% less clay and 10% more clay content. We have performed a sensitivity analysis using different clay content and found minimal effects on our results. Please see our response above (Response to Reviewer's #2) and articulated in the SI (text and figures).

How was the interaction between different sediment fractions dealt with in the model?

Response: As stated in the methods, we assume a homogenous mixed bed, with the said compositions (sand, silt, and clay). Clay is transported as cohesive and coarse silt, and sand as non-cohesive. All sediment types are available for erosion, with an initial sediment thickness of 7 m.

When is sediment within a mixed active layer top cell of the bed transported as cohesive or non-cohesive material?

Response: Both cohesive and non-cohesive sediments are actively (or available) transported within the active layer.

Which transitional coefficient did the authors use to delineate cohesive, from non-cohesive beds, e.g. the default, why?

This needs to be added and explained in the Methods Section.

Would the authors expect that this coefficient influences the robustness of their model predictions?

Response: During the first revision of the manuscript, we included the model setup files in a repository as requested by the reviewer. We do not fully understand what the reviewer is asking. However, from the methods, we re-iterate here that clay, which has an equivalent diameter of 20 microns in our model is delineated (14%) and transported as cohesive sediment. Any sediment with a diameter of more than 63 microns is delineated and transported as non-cohesive sediment. Since we are using a mixed bed. The fractions of 14% (clay, 20 microns), 43% (silt, 64 microns), and 43% (sand, 200 microns). The methods currently consist of all the above information.

We added a table with more information in the supplemental documents. We do not think the selection of these parameters or properties influences the robustness of the model. Moreover, the importance of the study lies in the differences in the results among the different simulations, thus, since all initial coefficients/conditions are the same between the simulations, the experiments remain robust.

3.b. This trend does not illustrate that the results derived (i.e., for the impact of SLR) from the model will be applicable to other Mixed-energy systems, see comment below: see point.10

Response: We understand and agree that the examples listed in Table 1 will not necessarily behave exactly like our conceptualized basin, because of differences in hypsometry, grain size distributions, tidal ranges, etc. and this acknowledgement is now clearly stated in the text. However, there is relevance to the systems listed in Table 1, because they similar basin lengths, tidal ranges and deep channels would tend to produce standing tidal wave regimes leading to similar hydrodynamics and ebb-dominated inlets. Moreover, these backbarrier systems have comparable ebb-tidal deltas and backbarriers consisting of marsh or well-developed tidal flats incised by tidal channels.

4. See comment above, and the impact of the ebb-tidal delta volume to wave forcing.

Previous studies have shown that ebb-tidal delta volume in mixed energy systems can be influenced by waves, e.g., Lenstra, Klaas JH, et al. "Cyclic channel-shoal dynamics at the Ameland inlet: the impact on waves, tides, and sediment transport." *Ocean Dynamics* 69 (2019): 409-425.

How would waves influence the predicted morphological development over SLR ?

Response: Lenstra et al's first figure and nomenclature in that publication are that from one of the present authors. Lenstra's paper focuses on time scales and processes of how sand bypasses an inlet through changes in the morphology and sediment transport patterns of the ebb-tidal delta. Indeed, Lenstra's paper expanded our understanding of the ebb-tidal delta breaching process and showed the importance of waves and tides. In our paper we are focusing on sea level induced volume changes and not detailed cyclic morphological changes, which is an important subject but beyond the scope of this paper.

5. I appreciate the revised version, leading to a much more streamlined linked between the sub-questions

Response: Thank you.

6. Although the system is a mixed-energy system, that does not necessarily mean results derived from this geometric model setup (presented in Table.1) are representative of other mixed energy systems.

Can derived results be applied to Frisian Island systems or other mixed energy systems along the US east coasts? I have major doubts about that. To make such a claim of the robustness of the results, we'd need to be evaluate varying geometry, bathymetry, initial sediment composition and wave/tide forcing ratios (within the mixed energy range),... which is beyond the scope of the current study.

Response: Yes, additional experiments are beyond the scope of this study, but please see above Response 3.b. which related our results to many other mixed-energy coastal systems due to similarities on the hydrodynamic regimes.

I agree the loss of fine sediments would potentially counteract salt marsh survival over sea level rise, this in itself would be a relevant finding for the Plum Island lagoon. I suggest that the authors explain why wave dynamics are less important at the current system, and then focus on the wetland fine sediment loss message.

Response: We are concentrating on hydrodynamics and sediment transport into and out of a backbarrier basin where wave energy is typically low due to limited fetch, particularly at mid to low tide. We agree that waves affect the morphology of the ebb-delta and processes of inlet sediment bypassing (cyclic behavior, see Lenstra et al); waves have a minor influence on ebb-tidal delta volume over centennial timescales.

However, extrapolating results to other barriers, is in my opinion not possible, without a significant investment in additional model scenarios.

Response: We feel that we have adequately addressed several comments related to this issue. Also, this study is about backbarrier basins and their evolution due to sea level rise.

7. As indicated by the authors, all the inlet's listed in Table.1 experience inlet sediment bypassing, pointing towards the importance of wave-induced long-shore transport around the ebb-tidal delta and thus wave processes. I understand that the authors' are modelling a conceptual basin, which I think is a valid approach, however the question remains whether all essential processes (specifically waves) have also been accounted for?

Response: Sediment transport within the basin and at the inlet is driven by tidal currents, dictating net transport directions and volumes. Waves influence the distribution of sand shoals, how they migrate, pattern of sand transport along the periphery of the ebb-delta, and to some extent "ebb-tidal delta breaching," but we are focusing on change to the basin with SLR. One of the results is an increase in tidal prism, which also causes an increase in tidal inlet cross sectional area and volume of the ebb delta, and both trends as predicted by the literature, and in our modeling.

8. I want to thank the authors for pointing me to the SI, showing the standing wave character and import of sand. However, SI.2 and 3 are model results; are there any indications e.g. field observations and statistical relationships by managing authorities that the predicted rates are in the right order of magnitude? Was there any validation carried out? for this or other comparable systems, e.g., scaling-relationships?

Response: Thank you for bringing this topic forward. There are three examples from the published literature that corroborate our findings. 1. *Ocean City Inlet, Maryland*. Hurricane breaching in 1933. As the inlet accessed the backbarrier lagoon and thus a greater tidal prism, the inlet deepened from 5 m in 1937 to > 14 m by 1995. Over the same period, the ebb delta volume enlarged to 10 million cubic meters. 2. *Barataria Basin in Louisiana*. Wetland loss increased over an 80 year period in Barataria Basin converted more than 1100 km² of wetlands to open water (14 km²/year; Couvillion et al 2011). Consequently, the combined cross-sectional areas of four inlets draining this basin increased by almost 70% and the ebb-tidal delta volumes increased by ~ 35% causing a seaward migration of the terminal lobe by 2 km (FitzGerald et al, 2018). 3. *East Frisian Islands*. Here, poldering decreased the area of the backbarrier > 60% vastly diminishing the tidal prism, which in turn resulted in a decrease in the cumulative widths of the inlets by >40% and cumulative lengthening of the barrier islands by 20%. The sand that contributed to the growth of barriers came from the ebb-tidal delta that decreased in size because of the decreasing tidal prism.

9. Which parameters have been used for the Baptist equation, more specifically, stem diameter, stem height, stem density and drag coefficient? This information is crucial for the reproducibility of the study.

Response: We have included the Table below in the Supplemental Information. We note that stem diameter is not a parameter in the Baptist equation.

SI Table 1. Vegetation parameters used in Baptist Equation

Parameter	Value
Stem Height [m]	0.5
Vegetation Density [stems/m]	0.05
Stem Density [stems/m ²]	185
Drag Coefficient	1
Alluvial Bed Roughness [m ^(1/2) /s]	60

10. I appreciate the updated abstract.

Unfortunately it remains unclear how sediment accretion was imposed, i.e. what sediment classes were added.

Line 413-430 and Line 450-460?

Did the authors simply increase the elevation on the most top-layers, including the active layer and all mixed sediment classes or did they add a specific size-fraction?

Is any portion of the accreted sediment part of sediment transport? Will this affect the resulting sediment balance, e.g., with marsh is converted to open water?

Response: We have revised the text in the methods for clarity and provide a response here as well. To simulate marsh vertical accretion, we modified the bathymetry of the model for areas that were occupied by vegetation. Mineral sediment was added (every 5 years) in the top (active) layer of the model, and the added sediment reflected the same composition as the underlying sediment (all sediment fractions). The added sediment is implicitly incorporated into the next sediment computations following sediment continuity and thus doesn't influence sediment imbalance. When a model cell containing marsh is eroded, the sediment in the cell enters the domain in the adjacent cells and transported per sediment transport module depended on the sediment type as previously described, and as stated in the methods.

Detailed suggestions:

I appreciate the work the authors have put into the revised version and the letter to the reviewer and responding to my suggestions.

However at several points it seems that resubmission might have been premature, still containing comments intended for within the author-group?

This could have been avoided.

Response: We apologize for this oversight.

Moreover, whereas most of the comments have been answered satisfactory.

Specific responses still did not provide sufficient information, which threatens the reproducibility:

e.g. Line 446: New Line 446, e.g. the Parteniades-Krone equation is used to fine sediment transport, which critical shear stress for erosion and erosion rate parameter has been used.

There should be a table in the Appendix, stating the chosen model setup?

Response: We created a table containing the chosen parameters, shown below and added to the Supplemental Information. Also per request by the reviewer in our first revision, we included simulation setup files in an online repository which has been active and available since the previous revision.

SI Table 2. Sediment Characteristics and other sediment and morphology relevant parameters

Category	Parameter	Values
Sediment characteristics	Sand median grain size (D50)	200 μm
	Coarse silt median grain size (D50)	64 μm
	Settling velocity (clay fraction)	0.25 mm/s (velocities correspond to D50s of 20 μm , respectively)

	Critical shear stress for erosion of consolidated mud (i.e., clay)	0.5 Pa
	Dry bed (or bulk) density of consolidated mud (i.e., silt and clay)	500 kg/m ³
	Dry bed (or bulk) density of sand	1600 kg/m ³
Sediment transport	Erosion parameter	0.0001 Kg/m²/s
	Current related reference concentration factor/transport magnitude factor	1 / 1
Morphology	Mobile bed composition	43% sand / 43% silt / 14% clay Sensitivity with 53% sand / 43% silt / 4% clay 33% sand / 43% silt / 24% clay
	Morphological scale factor (MorFac) for cold-front simulations	100
	Dry cell erosion factor	1

e.g. Line 463: how was the MEM model implement, see comment above what sediment classes have been added ?

Response: We did not use the MEM and as such we removed mention of MEM that from the method section. Please also see response to how accretion was implemented in a previous comment.

e.g. Line 471: again the specific parameter settings are missing which should be provided in some table in the appendix

Response: The requested table is shown above and was also included in the paper supplement.

REVIEWERS' COMMENTS

Reviewer #2 (Remarks to the Author):

The authors did a good job with the revision. All issues related to my previous reviews are resolved now. Therefore, I recommend publication of the paper.

Zheng Bing Wang

Reviewer #3 (Remarks to the Author):

I would like to thank the authors for doing a thorough job of responding to my earlier comments, including additional analyses.

Their detailed responses have resolved the concerns raised. I recommend publication.

REVIEWERS' COMMENTS

Reviewer #2 (Remarks to the Author):

The authors did a good job with the revision. All issues related to my previous reviews are resolved now. Therefore, I recommend publication of the paper.

Zheng Bing Wang

Response: Thank you

Reviewer #3 (Remarks to the Author):

I would like to thank the authors for doing a thorough job of responding to my earlier comments, including additional analyses.

Their detailed responses have resolved the concerns raised. I recommend publication.

Response: Thank you